# SELF-SUPERVISED REPRESENTATION LEARNING WITH RELATIVE PREDICTIVE CODING

**Yao-Hung Hubert Tsai**[1]**, Martin Q. Ma**[1]**, Muqiao Yang**[1]**,**
**Han Zhao**[23]**, Louis-Philippe Morency**[1]**, Ruslan Salakhutdinov**[1]
[1]Carnegie Mellon University, [2]D.E. Shaw & Co., [3] University of Illinois at Urbana-Champaign

## ABSTRACT

This paper introduces Relative Predictive Coding (RPC), a new contrastive representation learning objective that maintains a good balance among training stability, minibatch size sensitivity, and downstream task performance. The key to the success of RPC is two-fold. First, RPC introduces the relative parameters to regularize the objective for boundedness and low variance. Second, RPC contains no logarithm and exponential score functions, which are the main cause of training instability in prior contrastive objectives. We empirically verify the effectiveness of RPC on benchmark vision and speech self-supervised learning tasks. Lastly, we relate RPC with mutual information (MI) estimation, showing RPC can be used to estimate MI with low variance [1].

## 1 INTRODUCTION

Unsupervised learning has drawn tremendous attention recently because it can extract rich representations without label supervision. Self-supervised learning, a subset of unsupervised learning, learns representations by allowing the data to provide supervision (Devlin et al., 2018). Among its mainstream strategies, self-supervised contrastive learning has been successful in visual object recognition (He et al., 2020; Tian et al., 2019; Chen et al., 2020c), speech recognition (Oord et al., 2018; Rivière et al., 2020), language modeling (Kong et al., 2019), graph representation learning (Velickovic et al., 2019) and reinforcement learning (Kipf et al., 2019). The idea of self-supervised contrastive learning is to learn latent representations such that related instances (e.g., patches from the same image; defined as *positive* pairs) will have representations within close distance, while unrelated instances (e.g., patches from two different images; defined as *negative* pairs) will have distant representations (Arora et al., 2019).

Prior work has formulated the contrastive learning objectives as maximizing the divergence between the distribution of related and unrelated instances. In this regard, different divergence measurement often leads to different loss function design. For example, variational mutual information (MI) estimation (Poole et al., 2019) inspires Contrastive Predictive Coding (CPC) (Oord et al., 2018). Note that MI is also the KL-divergence between the distributions of related and unrelated instances (Cover & Thomas, 2012). While the choices of the contrastive learning objectives are abundant (Hjelm et al., 2018; Poole et al., 2019; Ozair et al., 2019), we point out that there are three challenges faced by existing methods.

The first challenge is the training stability, where an unstable training process with high variance may be problematic. For example, Hjelm et al. (2018); Tschannen et al. (2019); Tsai et al. (2020b) show that the contrastive objectives with large variance cause numerical issues and have a poor downstream performance with their learned representations. The second challenge is the sensitivity to minibatch size, where the objectives requiring a huge minibatch size may restrict their practical usage. For instance, SimCLRv2 (Chen et al., 2020c) utilizes CPC as its contrastive objective and reaches state-of-the-art performances on multiple self-supervised and semi-supervised benchmarks. Nonetheless, the objective is trained with a minibatch size of $8,192$, and this scale of training requires enormous computational power. The third challenge is the downstream task performance, which is the one that we would like to emphasize the most. For this reason, in most cases, CPC

---

[1]Project page: https://github.com/martinmamql/relative_predictive_coding

Table 1: Different contrastive learning objectives, grouped by measurements of distribution divergence. $P_{XY}$ represents the distribution of related samples (positively-paired), and $P_X P_Y$ represents the distribution of unrelated samples (negatively-paired). $f(x, y) \in \mathcal{F}$ for $\mathcal{F}$ being any class of functions $f : \mathcal{X} \times \mathcal{Y} \rightarrow \mathbb{R}$. †: Compared to $J_{\mathrm{CPC}}$ and $J_{\mathrm{RPC}}$, we empirically find $J_{\mathrm{WPC}}$ performs worse on complex real-world image datasets spanning CIFAR-10/-100 (Krizhevsky et al., 2009) and ImageNet (Russakovsky et al., 2015).

| Objective | Good Training Stability | Lower Minibatch Size Sensitivity | Good Downstream Performance |
|---|---|---|---|
| relating to KL-divergence between $P_{XY}$ and $P_X P_Y$: $J_{\mathrm{DV}}$ (Donsker & Varadhan, 1975), $J_{\mathrm{NWJ}}$ (Nguyen et al., 2010), and $J_{\mathrm{CPC}}$ (Oord et al., 2018) | | | |
| $J_{\mathrm{DV}}(X,Y) := \sup_{f \in \mathcal{F}} \mathbb{E}_{P_{XY}}[f(x,y)] - \log(\mathbb{E}_{P_X P_Y}[e^{f(x,y)}])$ | ✗ | ✓ | ✗ |
| $J_{\mathrm{NWJ}}(X,Y) := \sup_{f \in \mathcal{F}} \mathbb{E}_{P_{XY}}[f(x,y)] - \mathbb{E}_{P_X P_Y}[e^{f(x,y)-1}]$ | ✗ | ✓ | ✗ |
| $J_{\mathrm{CPC}}(X,Y) := \sup_{f \in \mathcal{F}} \mathbb{E}_{(x,y_1) \sim P_{XY}, \{y_j\}_{j=2}^N \sim P_Y} \left[ \log \left( e^{f(x,y_1)} / \frac{1}{N} \sum_{j=1}^N e^{f(x,y_j)} \right) \right]$ | ✓ | ✗ | ✓ |
| relating to JS-divergence between $P_{XY}$ and $P_X P_Y$: $J_{\mathrm{JS}}$ (Nowozin et al., 2016) | | | |
| $J_{\mathrm{JS}}(X,Y) := \sup_{f \in \mathcal{F}} \mathbb{E}_{P_{XY}}[-\log(1 + e^{-f(x,y)})] - \mathbb{E}_{P_X P_Y}[\log(1 + e^{f(x,y)})]$ | ✓ | ✓ | ✗ |
| relating to Wasserstein-divergence between $P_{XY}$ and $P_X P_Y$: $J_{\mathrm{WPC}}$ (Ozair et al., 2019), with $\mathcal{F}_\mathcal{L}$ denoting the space of 1-Lipschitz functions | | | |
| $J_{\mathrm{WPC}}(X,Y) := \sup_{f \in \mathcal{F}_\mathcal{L}} \mathbb{E}_{(x,y_1) \sim P_{XY}, \{y_j\}_{j=2}^N \sim P_Y} \left[ \log \left( e^{f(x,y_1)} / \frac{1}{N} \sum_{j=1}^N e^{f(x,y_j)} \right) \right]$ | ✓ | ✓ | ✗† |
| relating to $\chi^2$-divergence between $P_{XY}$ and $P_X P_Y$: $J_{\mathrm{RPC}}$ (ours) | | | |
| $J_{\mathrm{RPC}}(X,Y) := \sup_{f \in \mathcal{F}} \mathbb{E}_{P_{XY}}[f(x,y)] - \alpha \mathbb{E}_{P_X P_Y}[f(x,y)] - \frac{\beta}{2} \mathbb{E}_{P_{XY}}[f^2(x,y)] - \frac{\gamma}{2} \mathbb{E}_{P_X P_Y}[f^2(x,y)]$ | ✓ | ✓ | ✓ |

is the objective that we would adopt for contrastive representation learning, due to its favorable performance in downstream tasks (Tschannen et al., 2019; Baevski et al., 2020).

This paper presents a new contrastive representation learning objective: the Relative Predictive Coding (*RPC*), which attempts to achieve a good balance among these three challenges: training stability, sensitivity to minibatch size, and downstream task performance. At the core of RPC is the *relative parameters*, which are used to regularize RPC for its boundedness and low variance. From a modeling perspective, the relative parameters act as a $\ell_2$ regularization for RPC. From a statistical perspective, the relative parameters prevent RPC from growing to extreme values, as well as upper bound its variance. In addition to the relative parameters, RPC contains no logarithm and exponential, which are the main cause of the training instability for prior contrastive learning objectives (Song & Ermon, 2019).

To empirically verify the effectiveness of RPC, we consider benchmark self-supervised representation learning tasks, including visual object classification on CIFAR-10/-100 (Krizhevsky et al., 2009), STL-10 (Coates et al., 2011), and ImageNet (Russakovsky et al., 2015) and speech recognition on LibriSpeech (Panayotov et al., 2015). Comparing RPC to prior contrastive learning objectives, we observe a lower variance during training, a lower minibatch size sensitivity, and consistent performance improvement. Lastly, we also relate RPC with MI estimation, empirically showing that RPC can estimate MI with low variance.

## 2  PROPOSED METHOD

This paper presents a new contrastive representation learning objective - the Relative Predictive Coding (RPC). At a high level, RPC 1) introduces the relative parameters to regularize the objective for boundedness and low variance; and 2) achieves a good balance among the three challenges in the contrastive representation learning objectives: training stability, sensitivity to minibatch size, and downstream task performance. We begin by describing prior contrastive objectives along with their limitations on the three challenges in Section 2.1. Then, we detail our presented objective and its modeling benefits in Section 2.2. An overview of different contrastive learning objectives is provided in Table 1. We defer all the proofs in Appendix.

**Notation**  We use an uppercase letter to denote a random variable (e.g., $X$), a lower case letter to denote the outcome of this random variable (e.g., $x$), and a calligraphy letter to denote the sample space of this random variable (e.g., $\mathcal{X}$). Next, if the samples $(x, y)$ are related (or positively-paired), we refer $(x, y) \sim P_{XY}$ with $P_{XY}$ being the joint distribution of $X \times Y$. If the samples $(x, y)$ are unrelated (negatively-paired), we refer $(x, y) \sim P_X P_Y$ with $P_X P_Y$ being the product of marginal distributions over $X \times Y$. Last, we define $f \in \mathcal{F}$ for $\mathcal{F}$ being any class of functions $f : \mathcal{X} \times \mathcal{Y} \rightarrow \mathbb{R}$.

### 2.1  PRELIMINARY

Contrastive representation learning encourages the *contrastiveness* between the positive and the negative pairs of the representations from the related data $X$ and $Y$. Specifically, when sampling a pair

of representations $(x, y)$ from their joint distribution $((x, y) \sim P_{XY})$, this pair is defined as a positive pair; when sampling from the product of marginals $((x, y) \sim P_X P_Y)$, this pair is defined as a negative pair. Then, Tsai et al. (2020b) formalizes this idea such that the contrastiveness of the representations can be measured by the divergence between $P_{XY}$ and $P_X P_Y$, where higher divergence suggests better contrastiveness. To better understand prior contrastive learning objectives, we categorize them in terms of different divergence measurements between $P_{XY}$ and $P_X P_Y$, with their detailed objectives presented in Table 1.

We instantiate the discussion using Contrastive Predictive Coding (Oord et al., 2018, $J_{\mathrm{CPC}}$), which is a lower bound of $D_{\mathrm{KL}}(P_{XY} \| P_X P_Y)$ with $D_{\mathrm{KL}}$ referring to the KL-divergence:

$$J_{\mathrm{CPC}}(X, Y) := \sup_{f \in \mathcal{F}} \mathbb{E}_{(x, y_1) \sim P_{XY}, \{y_j\}_{j=2}^N \sim P_Y} \left[ \log \frac{e^{f(x, y_1)}}{\frac{1}{N} \sum_{j=1}^N e^{f(x, y_j)}} \right]. \tag{1}$$

Then, Oord et al. (2018) presents to maximize $J_{\mathrm{CPC}}(X, Y)$, so that the learned representations $X$ and $Y$ have high contrastiveness. We note that $J_{\mathrm{CPC}}$ has been commonly used in many recent self-supervised representation learning frameworks (He et al., 2020; Chen et al., 2020b), where they constrain the function to be $f(x, y) = \mathrm{cosine}(x, y)$ with $\mathrm{cosine}(\cdot)$ being cosine similarity. Under this function design, maximizing $J_{\mathrm{CPC}}$ leads the representations of related pairs to be close and representations of unrelated pairs to be distant.

The category of modeling $D_{\mathrm{KL}}(P_{XY} \| P_X P_Y)$ also includes the Donsker-Varadhan objective ($J_{\mathrm{DV}}$ (Donsker & Varadhan, 1975; Belghazi et al., 2018)) and the Nguyen-Wainright-Jordan objective ($J_{\mathrm{NWJ}}$ (Nguyen et al., 2010; Belghazi et al., 2018)), where Belghazi et al. (2018); Tsai et al. (2020b) show that $J_{\mathrm{DV}}(X, Y) = J_{\mathrm{NWJ}}(X, Y) = D_{\mathrm{KL}}(P_{XY} \| P_X P_Y)$. The other divergence measurements considered in prior work are $D_{\mathrm{JS}}(P_{XY} \| P_X P_Y)$ (with $D_{\mathrm{JS}}$ referring to the Jenson-Shannon divergence) and $D_{\mathrm{Wass}}(P_{XY} \| P_X P_Y)$ (with $D_{\mathrm{Wass}}$ referring to the Wasserstein-divergence). The instance of modeling $D_{\mathrm{JS}}(P_{XY} \| P_X P_Y)$ is the Jensen-Shannon f-GAN objective $\left(J_{\mathrm{JS}}$ (Nowozin et al., 2016; Hjelm et al., 2018)$\right)$, where $J_{\mathrm{JS}}(X, Y) = 2\left(D_{\mathrm{JS}}(P_{XY} \| P_X P_Y) - \log 2\right)$.[2] The instance of modeling $D_{\mathrm{Wass}}(P_{XY} \| P_X P_Y)$ is the Wasserstein Predictive Coding $\left(J_{\mathrm{WPC}}$ (Ozair et al., 2019)$\right)$, where $J_{\mathrm{WPC}}(X, Y)$ modifies $J_{\mathrm{CPC}}(X, Y)$ objective (equation 1) by searching the function from $\mathcal{F}$ to $\mathcal{F}_{\mathcal{L}}$. $\mathcal{F}_{\mathcal{L}}$ denotes any class of 1-Lipschitz continuous functions from $(\mathcal{X} \times \mathcal{Y})$ to $\mathbb{R}$, and thus $\mathcal{F}_{\mathcal{L}} \subset \mathcal{F}$. Ozair et al. (2019) shows that $J_{\mathrm{WPC}}(X, Y)$ is the lower bound of both $D_{\mathrm{KL}}(P_{XY} \| P_X P_Y)$ and $D_{\mathrm{Wass}}(P_{XY} \| P_X P_Y)$. See Table 1 for all the equations. To conclude, the contrastive representation learning objectives are unsupervised representation learning methods that maximize the distribution divergence between $P_{XY}$ and $P_X P_Y$. The learned representations cause high contrastiveness, and recent work (Arora et al., 2019; Tsai et al., 2020a) theoretically show that highly-contrastive representations could improve the performance on downstream tasks.

After discussing prior contrastive representation learning objectives, we point out three challenges in their practical deployments: training stability, sensitivity to minibatch training size, and downstream task performance. In particular, the three challenges can hardly be handled well at the same time, where we highlight the conclusions in Table 1. ***Training Stability:*** The training stability highly relates to the variance of the objectives, where Song & Ermon (2019) shows that $J_{\mathrm{DV}}$ and $J_{\mathrm{NWJ}}$ exhibit inevitable high variance due to their inclusion of exponential function. As pointed out by Tsai et al. (2020b), $J_{\mathrm{CPC}}$, $J_{\mathrm{WPC}}$, and $J_{\mathrm{JS}}$ have better training stability because $J_{\mathrm{CPC}}$ and $J_{\mathrm{WPC}}$ can be realized as a multi-class classification task and $J_{\mathrm{JS}}$ can be realized as a binary classification task. The cross-entropy loss adopted in $J_{\mathrm{CPC}}$, $J_{\mathrm{WPC}}$, and $J_{\mathrm{JS}}$ is highly-optimized and stable in existing optimization package (Abadi et al., 2016; Paszke et al., 2019). ***Sensitivity to minibatch training size:*** Among all the prior contrastive representation learning methods, $J_{\mathrm{CPC}}$ is known to be sensitive to the minibatch training size (Ozair et al., 2019). Taking a closer look at equation 1, $J_{\mathrm{CPC}}$ deploys an instance selection such that $y_1$ should be selected from $\{y_1, y_2, \cdots, y_N\}$, with $(x, y_1) \sim P_{XY}$, $(x, y_{j>1}) \sim P_X P_Y$ with $N$ being the minibatch size. Previous work (Poole et al., 2019; Song & Ermon, 2019; Chen et al., 2020b; Caron et al., 2020) showed that a large $N$ results in a more challenging instance selection and forces $J_{\mathrm{CPC}}$ to have a better contrastiveness of $y_1$ (related instance for $x$) against $\{y_j\}_{j=2}^N$ (unrelated instance for $x$). $J_{\mathrm{DV}}$, $J_{\mathrm{NWJ}}$, and $J_{\mathrm{JS}}$ do not consider

---

[2]$J_{\mathrm{JS}}(X, Y)$ achieves its supreme value when $f^*(x, y) = \log(p(x, y)/p(x)p(y))$ (Tsai et al., 2020b). Plug-in $f^*(x, y)$ into $J_{\mathrm{JS}}(X, Y)$, we can conclude $J_{\mathrm{JS}}(X, Y) = 2(D_{\mathrm{JS}}(P_{XY} \| P_X P_Y) - \log 2)$.

the instance selection, and $J_{\text{WPC}}$ reduces the minibatch training size sensitivity by enforcing 1-Lipschitz constraint. ***Downstream Task Performance:*** The downstream task performance is what we care the most among all the three challenges. $J_{\text{CPC}}$ has been the most popular objective as it manifests superior performance over the other alternatives (Tschannen et al., 2019; Tsai et al., 2020b;a). We note that although $J_{\text{WPC}}$ shows better performance on Omniglot (Lake et al., 2015) and CelebA (Liu et al., 2015) datasets, we empirically find it not generalizing well to CIFAR-10/-100 (Krizhevsky et al., 2009) and ImageNet (Russakovsky et al., 2015).

## 2.2 RELATIVE PREDICTIVE CODING

In this paper, we present Relative Predictive Coding (RPC), which achieves a good balance among the three challenges mentioned above:

$$J_{\text{RPC}}(X,Y) := \sup_{f \in \mathcal{F}} \mathbb{E}_{P_{XY}}[f(x,y)] - \alpha \mathbb{E}_{P_X P_Y}[f(x,y)] - \frac{\beta}{2}\mathbb{E}_{P_{XY}}\left[f^2(x,y)\right] - \frac{\gamma}{2}\mathbb{E}_{P_X P_Y}\left[f^2(x,y)\right],$$

(2)

where $\alpha > 0$, $\beta > 0$, $\gamma > 0$ are hyper-parameters and we define them as *relative parameters*. Intuitively, $J_{\text{RPC}}$ contains no logarithm or exponential, potentially preventing unstable training due to numerical issues. Now, we discuss the roles of $\alpha, \beta, \gamma$. At a first glance, $\alpha$ acts to discourage the scores of $P_{XY}$ and $P_X P_Y$ from being close, and $\beta/\gamma$ acts as a $\ell_2$ regularization coefficient to stop $f$ from becoming large. For a deeper analysis, the relative parameters act to regularize our objective for boundedness and low variance. To show this claim, we first present the following lemma:

**Lemma 1 (Optimal Solution for $J_{\text{RPC}}$)** *Let $r(x,y) = \frac{p(x,y)}{p(x)p(y)}$ be the density ratio. $J_{\text{RPC}}$ has the optimal solution $f^*(x,y) = \frac{r(x,y) - \alpha}{\beta\, r(x,y) + \gamma} := r_{\alpha,\beta,\gamma}(x,y)$ with $-\frac{\alpha}{\gamma} \leq r_{\alpha,\beta,\gamma} \leq \frac{1}{\beta}$.*

Lemma 1 suggests that $J_{\text{RPC}}$ achieves its supreme value at the ratio $r_{\alpha,\beta,\gamma}(x,y)$ indexed by the relative parameters $\alpha, \beta, \gamma$ (i.e., we term $r_{\alpha,\beta,\gamma}(x,y)$ as the relative density ratio). We note that $r_{\alpha,\beta,\gamma}(x,y)$ is an increasing function w.r.t. $r(x,y)$ and is nicely bounded even when $r(x,y)$ is large. We will now show that the bounded $r_{\alpha,\beta,\gamma}$ suggests the empirical estimation of $J_{\text{RPC}}$ has boundeness and low variance. In particular, let $\{x_i, y_i\}_{i=1}^n$ be $n$ samples drawn uniformly at random from $P_{XY}$ and $\{x'_j, y'_j\}_{j=1}^m$ be $m$ samples drawn uniformly at random from $P_X P_Y$. Then, we use neural networks to empirically estimate $J_{\text{RPC}}$ as $\hat{J}_{\text{RPC}}^{m,n}$:

**Definition 1 ($\hat{J}_{\text{RPC}}^{m,n}$, empirical estimation of $J_{\text{RPC}}$)** *We parametrize $f$ via a family of neural networks $\mathcal{F}_\Theta := \{f_\theta : \theta \in \Theta \subseteq \mathbb{R}^d\}$ where $d \in \mathbb{N}$ and $\Theta$ is compact. Then, $\hat{J}_{\text{RPC}}^{m,n} = \sup_{f_\theta \in \mathcal{F}_\Theta} \frac{1}{n}\sum_{i=1}^n f_\theta(x_i, y_i) - \frac{1}{m}\sum_{j=1}^m \alpha f_\theta(x'_j, y'_j) - \frac{1}{n}\sum_{i=1}^n \frac{\beta}{2} f_\theta^2(x_i, y_i) - \frac{1}{m}\sum_{j=1}^m \frac{\gamma}{2} f_\theta^2(x'_j, y'_j)$.*

**Proposition 1 (Boundedness of $\hat{J}_{\text{RPC}}^{m,n}$, informal)** $0 \leq J_{\text{RPC}} \leq \frac{1}{2\beta} + \frac{\alpha^2}{2\gamma}$. *Then, with probability at least $1 - \delta$, $|J_{\text{RPC}} - \hat{J}_{\text{RPC}}^{m,n}| = O(\sqrt{\frac{d + \log(1/\delta)}{n'}})$, where $n' = \min\{n, m\}$.*

**Proposition 2 (Variance of $\hat{J}_{\text{RPC}}^{m,n}$, informal)** *There exist universal constants $c_1$ and $c_2$ that depend only on $\alpha, \beta, \gamma$, such that $\text{Var}[\hat{J}_{\text{RPC}}^{m,n}] = O\left(\frac{c_1}{n} + \frac{c_2}{m}\right)$.*

From the two propositions, when $m$ and $n$ are large, i.e., the sample sizes are large, $\hat{J}_{\text{RPC}}^{m,n}$ is bounded, and its variance vanishes to 0. First, the boundedness of $\hat{J}_{\text{RPC}}^{m,n}$ suggests $\hat{J}_{\text{RPC}}^{m,n}$ will not grow to extremely large or small values. Prior contrastive learning objectives with good training stability (e.g., $J_{\text{CPC}}/J_{\text{JS}}/J_{\text{WPC}}$) also have the boundedness of their objective values. For instance, the empirical estimation of $J_{\text{CPC}}$ is less than $\log N$ (equation 1) (Poole et al., 2019). Nevertheless, $J_{\text{CPC}}$ often performs the best only when minibatch size is large, and empirical performances of $J_{\text{JS}}$ and $J_{\text{WPC}}$ are not as competitive as $J_{\text{CPC}}$. Second, the upper bound of the variance implies the training of $\hat{J}_{\text{RPC}}^{m,n}$ can be stable, and in practice we observe a much smaller value than the stated upper bound. On the contrary, Song & Ermon (2019) shows that the empirical estimations of $J_{\text{DV}}$ and $J_{\text{NWJ}}$ exhibit inevitable variances that grow exponentially with the true $D_{\text{KL}}(P_{XY}\|P_X P_Y)$.

Lastly, similar to prior contrastive learning objective that are related to distribution divergence measurement, we associate $J_{\text{RPC}}$ with the Chi-square divergence $D_{\chi^2}(P_{XY} \| P_X P_Y) = $

$\mathbb{E}_{P_X P_Y}[r^2(x,y)] - 1$ (Nielsen & Nock, 2013). The derivations are provided in Appendix. By having $P' = \frac{\beta}{\beta+\gamma}P_{XY} + \frac{\gamma}{\beta+\gamma}P_X P_Y$ as the mixture distribution of $P_{XY}$ and $P_X P_Y$, we can rewrite $J_{\mathrm{RPC}}(X,Y)$ as $J_{\mathrm{RPC}}(X,Y) = \frac{\beta+\gamma}{2}\mathbb{E}_{P'}[r^2_{\alpha,\beta,\gamma}(x,y)]$. Hence, $J_{\mathrm{RPC}}$ can be regarded as a generalization of $D_{\chi^2}$ with the relative parameters $\alpha, \beta, \gamma$, where $D_{\chi^2}$ can be recovered from $J_{\mathrm{RPC}}$ by specializing $\alpha = 0$, $\beta = 0$ and $\gamma = 1$ (e.g., $D_{\chi^2} = 2J_{\mathrm{RPC}}|_{\alpha=\beta=0,\gamma=1} - 1$). Note that $J_{\mathrm{RPC}}$ may not be a formal divergence measure with arbitrary $\alpha, \beta, \gamma$.

## 3 EXPERIMENTS

We provide an overview of the experimental section. First, we conduct benchmark self-supervised representation learning tasks spanning visual object classification and speech recognition. This set of experiments are designed to discuss the three challenges of the contrastive representation learning objectives: downstream task performance (Section 3.1), training stability (Section 3.2), and mini-batch size sensitivity (Section 3.3). We also provide an ablation study on the choices of the relative parameters in $J_{\mathrm{RPC}}$ (Section 3.4). On these experiments we found that $J_{\mathrm{RPC}}$ achieves a lower variance during training, a lower batch size insensitivity, and consistent performance improvement. Second, we relate $J_{\mathrm{RPC}}$ with mutual information (MI) estimation (Section 3.5). The connection is that MI is an average statistic of the density ratio, and we have shown that the optimal solution of $J_{\mathrm{RPC}}$ is the relative density ratio (see Lemma 1). Thus we could estimate MI using the density ratio transformed from the optimal solution of $J_{\mathrm{RPC}}$. On these two sets of experiments, we fairly compare $J_{\mathrm{RPC}}$ with other contrastive learning objectives. Particularly, across different objectives, we fix the network, learning rate, optimizer, and batch size (we use the default configurations suggested by the original implementations from Chen et al. (2020c), Rivière et al. (2020) and Tsai et al. (2020b).) The only difference will be the objective itself. In what follows, we perform the first set of experiments. We defer experimental details in the Appendix.

**Datasets.** For the visual objective classification, we consider CIFAR-10/-100 (Krizhevsky et al., 2009), STL-10 (Coates et al., 2011), and ImageNet (Russakovsky et al., 2015). CIFAR-10/-100 and ImageNet contain labeled images only, while STL-10 contains labeled and unlabeled images. For the speech recognition, we consider LibriSpeech-100h (Panayotov et al., 2015) dataset, which contains 100 hours of 16kHz English speech from 251 speakers with 41 types of phonemes.

**Training and Evaluation Details.** For the vision experiments, we follow the setup from SimCLRv2 (Chen et al., 2020c), which considers visual object recognition as its downstream task. For the speech experiments, we follow the setup from prior work (Oord et al., 2018; Rivière et al., 2020), which consider phoneme classification and speaker identification as the downstream tasks. Then, we briefly discuss the training and evaluation details into three modules: 1) related and unrelated data construction, 2) pre-training, and 3) fine-tuning and evaluation. For more details, please refer to Appendix or the original implementations.

▷ *Related and Unrelated Data Construction.* In the vision experiment, we construct the related images by applying different augmentations on the same image. Hence, when $(x,y) \sim P_{XY}$, $x$ and $y$ are the same image with different augmentations. The unrelated images are two randomly selected samples. In the speech experiment, we define the current latent feature (feature at time $t$) and the future samples (samples at time $> t$) as related data. In other words, the feature in the latent space should contain information that can be used to infer future time steps. A latent feature and randomly selected samples would be considered as unrelated data.

▷ *Pre-training.* The pre-training stage refers to the self-supervised training by a contrastive learning objective. Our training objective is defined in Definition 1, where we use neural networks to parametrize the function using the constructed related and unrelated data. Convolutional neural networks are used for vision experiments. Transformers (Vaswani et al., 2017) and LSTMs (Hochreiter & Schmidhuber, 1997) are used for speech experiments.

▷ *Fine-tuning and Evaluation.* After the pre-training stage, we fix the parameters in the pre-trained networks and add a small fine-tuning network on top of them. Then, we fine-tune this small network with the downstream labels in the data's training split. For the fine-tuning network, both vision and speech experiments consider multi-layer perceptrons. Last, we evaluate the fine-tuned representations on the data's test split. We would like to point out that we do not normalize the hidden representations encoded by the pre-training neural network for loss calculation. This hidden nor-

Table 2: Top-1 accuracy (%) for visual object recognition results. $J_{DV}$ and $J_{NWJ}$ are not reported on ImageNet due to numerical instability. ResNet depth, width and Selective Kernel (SK) configuration for each setting are provided in ResNet depth+width+SK column. A slight drop of $J_{CPC}$ performance compared to Chen et al. (2020c) is because we only train for 100 epochs rather than 800 due to the fact that running 800 epochs uninterruptedly on cloud TPU is very expensive. Also, we did not employ a memory buffer (He et al., 2020) to store negative samples. We and we did not employ a memory buffer. We also provide the results from fully supervised models as a comparison (Chen et al., 2020b;c). Fully supervised training performs worse on STL-10 because it does not employ the unlabeled samples in the dataset (Löwe et al., 2019).

| Dataset | ResNet Depth+Width+SK | Self-supervised | | | | | | Supervised |
| --- | --- | --- | --- | --- | --- | --- | --- | --- |
| | | $J_{DV}$ | $J_{NWJ}$ | $J_{JS}$ | $J_{WPC}$ | $J_{CPC}$ | $J_{RPC}$ | |
| CIFAR-10 | $18 + 1\times$ + No SK | 91.10 | 90.54 | 83.55 | 80.02 | 91.12 | **91.46** | 93.12 |
| CIFAR-10 | $50 + 1\times$ + No SK | 92.23 | 92.67 | 87.34 | 85.93 | 93.42 | **93.57** | 95.70 |
| CIFAR-100 | $18 + 1\times$ + No SK | 77.10 | 77.27 | 74.02 | 72.16 | 77.36 | **77.98** | 79.11 |
| CIFAR-100 | $50 + 1\times$ + No SK | 79.02 | 78.52 | 75.31 | 73.23 | 79.31 | **79.89** | 81.20 |
| STL-10 | $50 + 1\times$ + No SK | 82.25 | 81.17 | 79.07 | 76.50 | 83.40 | **84.10** | 71.40 |
| ImageNet | $50 + 1\times$ + SK | - | - | 66.21 | 62.10 | 73.48 | **74.43** | 78.50 |
| ImageNet | $152 + 2\times$ + SK | - | - | 71.12 | 69.51 | 77.80 | **78.40** | 80.40 |

Table 3: Accuracy (%) for LibriSpeech-100h phoneme and speaker classification results. We also provide the results from fully supervised model as a comparison (Oord et al., 2018).

| Task Name | Self-supervised | | | | Supervised |
| --- | --- | --- | --- | --- | --- |
| | $J_{CPC}$ | $J_{DV}$ | $J_{NWJ}$ | $J_{RPC}$ | |
| Phoneme classification | 64.6 | 61.27 | 62.09 | **69.39** | 74.6 |
| Speaker classification | 97.4 | 95.36 | 95.89 | **97.68** | 98.5 |

malization technique is widely applied (Tian et al., 2019; Chen et al., 2020b;c) to stabilize training and increase performance for prior objectives, but we find it unnecessary in $J_{RPC}$.

## 3.1 DOWNSTREAM TASK PERFORMANCES ON VISION AND SPEECH

For the downstream task performance in the vision domain, we test the proposed $J_{RPC}$ and other contrastive learning objectives on CIFAR-10/-100 (Krizhevsky et al., 2009), STL-10 (Coates et al., 2011), and ImageNet ILSVRC-2012 (Russakovsky et al., 2015). Here we report the best performances $J_{RPC}$ can get on each dataset (we include experimental details in A.7.) Table 2 shows that the proposed $J_{RPC}$ outperforms other objectives on all datasets. Using $J_{RPC}$ on the largest network (ResNet with depth of 152, channel width of 2 and selective kernels), the performance jumps from 77.80% of $J_{CPC}$ to 78.40% of $J_{RPC}$.

Regarding speech representation learning, the downstream performance for phoneme and speaker classification are shown in Table 3 (we defer experimental details in Appendix A.9.) Compared to $J_{CPC}$, $J_{RPC}$ improves the phoneme classification results with 4.8 percent and the speaker classification results with 0.3 percent, which is closer to the fully supervised model. Overall, the proposed $J_{RPC}$ performs better than other unsupervised learning objectives on both phoneme classification and speaker classification tasks.

## 3.2 TRAINING STABILITY

We provide empirical training stability comparisons on $J_{DV}$, $J_{NWJ}$, $J_{CPC}$ and $J_{RPC}$ by plotting the values of the objectives as the training step increases. We apply the four objectives to the SimCLRv2 framework and train on the CIFAR-10 dataset. All setups of training are exactly the same except the objectives. From our experiments, $J_{DV}$ and $J_{NWJ}$ soon explode to NaN and disrupt training (shown as early stopping in Figure 1a; extremely large values are not plotted due to scale constraints). On the other hand, $J_{RPC}$ and $J_{CPC}$ has low variance, and both enjoy stable training. As a result, performances using the representation learned from unstable $J_{DV}$ and $J_{NWJ}$ suffer in downstream task, while representation learned by $J_{RPC}$ and $J_{CPC}$ work much better.

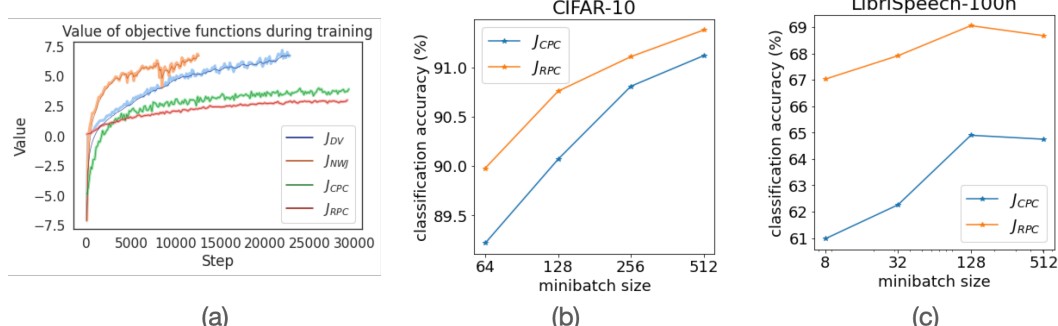

Figure 1: (a) Empirical values of $J_{\mathrm{DV}}$, $J_{\mathrm{NWJ}}$, $J_{\mathrm{CPC}}$ and $J_{\mathrm{RPC}}$ performing visual object recognition on CIFAR-10. $J_{\mathrm{DV}}$ and $J_{\mathrm{NWJ}}$ soon explode to NaN values and stop the training (shown as early stopping in the figure), while $J_{\mathrm{CPC}}$ and $J_{\mathrm{RPC}}$ are more stable. Performance comparison of $J_{\mathrm{CPC}}$ and $J_{\mathrm{RPC}}$ on (b) CIFAR-10 and (c) LibriSpeech-100h with different minibatch sizes, showing that the performance of $J_{\mathrm{RPC}}$ is less sensitive to minibatch size change compared to $J_{\mathrm{CPC}}$.

### 3.3 MINIBATCH SIZE SENSITIVITY

We then provide the analysis on the effect of minibatch size on $J_{\mathrm{RPC}}$ and $J_{\mathrm{CPC}}$, since $J_{\mathrm{CPC}}$ is known to be sensitive to minibatch size (Poole et al., 2019). We train SimCLRv2 (Chen et al., 2020c) on CIFAR-10 and the model from Rivière et al. (2020) on LibriSpeech-100h using $J_{\mathrm{RPC}}$ and $J_{\mathrm{CPC}}$ with different minibatch sizes. The settings of relative parameters are the same as Section 3.2. From Figure 1b and 1c, we can observe that both $J_{\mathrm{RPC}}$ and $J_{\mathrm{CPC}}$ achieve their optimal performance at a large minibatch size. However, when the minibatch size decreases, the performance of $J_{\mathrm{CPC}}$ shows higher sensitivity and suffers more when the number of minibatch samples is small. The result suggests that the proposed method might be less sensitive to the change of minibatch size compared to $J_{\mathrm{CPC}}$ given the same training settings.

### 3.4 EFFECT OF RELATIVE PARAMETERS

We study the effect of different combinations of relative parameters in $J_{\mathrm{RPC}}$ by comparing downstream performances on visual object recognition. We train SimCLRv2 on CIFAR-10 with different combinations of $\alpha, \beta$ and $\gamma$ in $J_{\mathrm{RPC}}$ and fix all other experimental settings. We choose $\alpha \in \{0, 0.001, 1.0\}, \beta \in \{0, 0.001, 1.0\}, \gamma \in \{0, 0.001, 1.0\}$ and we report the best performances under each combination of $\alpha, \beta$, and $\gamma$. From Figure 2, we first observe that $\alpha > 0$ has better downstream performance than $\alpha = 0$ when $\beta$ and $\gamma$ are fixed. This observation is as expected, since $\alpha > 0$ encourages representations of related and unrelated samples to be pushed away. Then, we find that a small but nonzero $\beta$ ($\beta = 0.001$) and a large $\gamma$ ($\gamma = 1.0$) give the best performance compared to other combinations. Since $\beta$ and $\gamma$ serve as the coefficients of $\ell_2$ regularization, the results imply that the regularization is a strong and sensitive factor that will influence the performance. The results here are not as competitive as Table 2 because the CIFAR-10 result reported in Table 2 is using a set of relative parameters ($\alpha = 1.0, \beta = 0.005, \gamma = 1.0$) that is different from the combinations in this subsection. Also, we use quite different ranges of $\gamma$ on ImageNet (see A.7 for details.) In conclusion, we find empirically that a non-zero $\alpha$, a small $\beta$ and a large $\gamma$ will lead to the optimal representation for the downstream task on CIFAR-10.

### 3.5 RELATION TO MUTUAL INFORMATION ESTIMATION

The presented approach also closely relates to mutual information estimation. For random variables $X$ and $Y$ with joint distribution $P_{XY}$ and product of marginals $P_X P_Y$, the mutual information is defined as $I(X; Y) = D_{\mathrm{KL}}(P_{XY} \| P_X P_Y)$. Lemma 1 states that given optimal solution $f^*(x, y)$ of $J_{\mathrm{RPC}}$, we can get the density ratio $r(x, y) := p(x, y)/p(x)p(y)$ as $r(x, y) = \frac{\gamma/\beta + \alpha}{1 - \beta f^*(x, y)} - \frac{\gamma}{\beta}$. We can empirically estimate $\hat{r}(x, y)$ from the estimated $\hat{f}(x, y)$ via this transformation, and use $\hat{r}(x, y)$ to estimate mutual information (Tsai et al., 2020b). Specifically, $I(X; Y) \approx \frac{1}{n} \sum_{i=1}^{n} \log \hat{r}(x_i, y_i)$ with $(x_i, y_i) \sim P_{X,Y}^{\otimes n}$, where $P_{X,Y}^{\otimes n}$ is the uniformly sampled empirical distribution of $P_{X,Y}$.

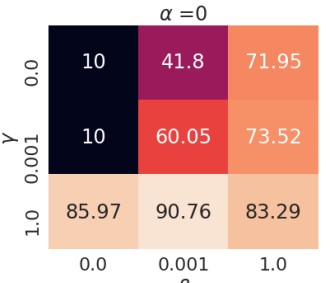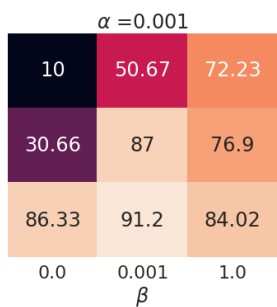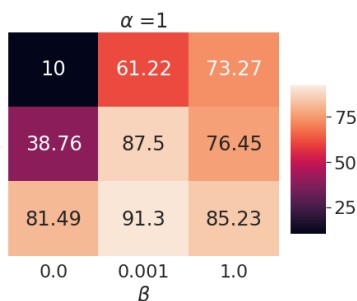

Figure 2: Heatmaps of downstream task performance on CIFAR-10, using different $\alpha$, $\beta$ and $\gamma$ in the $J_{\mathrm{RPC}}$. We conclude that a nonzero $\alpha$, a small $\beta$ ($\beta = 0.001$) and a large $\gamma(\gamma = 1.0)$ are crucial for better performance.

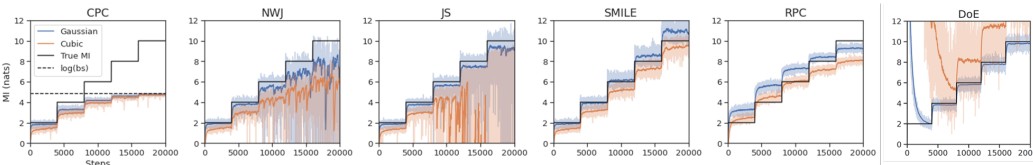

Figure 3: Mutual information estimation performed on 20-d correlated Gaussian distribution, with the correlation increasing each 4K steps. $J_{\mathrm{RPC}}$ exhibits smaller variance than SMILE and DoE, and smaller bias than $J_{\mathrm{CPC}}$.

We follow prior work (Poole et al., 2019; Song & Ermon, 2019; Tsai et al., 2020b) for the experiments. We consider $X$ and $Y$ as two 20-dimensional Gaussians with correlation $\rho$, and our goal is to estimate the mutual information $I(X; Y)$. Then, we perform a cubic transformation on $y$ so that $y \mapsto y^3$. The first task is referred to as **Gaussian** task and the second is referred to as **Cubic** task, where both have the ground truth $I(X; Y) = -10\log\left(1 - \rho^2\right)$. The models are trained on $20,000$ steps with $I(X; Y)$ starting at 2 and increased by 2 per $4,000$ steps. Our method is compared with baseline methods $J_{\mathrm{CPC}}$ (Oord et al., 2018), $J_{\mathrm{NWJ}}$ (Nguyen et al., 2010), $J_{\mathrm{JS}}$ (Nowozin et al., 2016), SMILE (Song & Ermon, 2019) and Difference of Entropies (DoE) (McAllester & Stratos, 2020). All approaches use the same network design, learning rate, optimizer and minibatch size for a fair comparison. First, we observe $J_{\mathrm{CPC}}$ (Oord et al., 2018) has the smallest variance, while it exhibits a large bias (the estimated mutual information from $J_{\mathrm{CPC}}$ has an upper bound $\log(\mathrm{batch\,size})$). Second, $J_{\mathrm{NWJ}}$ (Nguyen et al., 2010) and $J_{\mathrm{JSD}}$ (Poole et al., 2019) have large variances, especially in the Cubic task. Song & Ermon (2019) pointed out the limitations of $J_{\mathrm{CPC}}$, $J_{\mathrm{NWJ}}$, and $J_{\mathrm{JSD}}$, and developed the SMILE method, which clips the value of the estimated density function to reduce the variance of the estimators. DoE (McAllester & Stratos, 2020) is neither a lower bound nor a upper bound of mutual information, but can achieve accurate estimates when underlying mutual information is large. $J_{\mathrm{RPC}}$ exhibits comparable bias and lower variance compared to the SMILE method, and is more stable than the DoE method. We would like to highlight our method's low-variance property, where we neither clip the values of the estimated density ratio nor impose an upper bound of our estimated mutual information.

## 4 RELATED WORK

As a subset of unsupervised representation learning, self-supervised representation learning (SSL) adopts self-defined signals as supervision and uses the learned representation for downstream tasks, such as object detection and image captioning (Liu et al., 2020). We categorize SSL work into two groups: when the signal is the input's hidden property or the corresponding view of the input. For the first group, for example, Jigsaw puzzle (Noroozi & Favaro, 2016) shuffles the image patches and defines the SSL task for predicting the shuffled positions of the image patches. Other instances are Predicting Rotations (Gidaris et al., 2018) and Shuffle & Learn (Misra et al., 2016). For the second group, the SSL task aims at modeling the co-occurrence of multiple views of data, via the contrastive or the predictive learning objectives (Tsai et al., 2020a). The predictive objectives encourage reconstruction from one view of the data to the other, such as predicting the lower part of an image from

its upper part (ImageGPT by Chen et al. (2020a)). Comparing the contrastive with predictive learning approaches, Tsai et al. (2020a) points out that the former requires less computational resources for a good performance but suffers more from the over-fitting problem.

Theoretical analysis (Arora et al., 2019; Tsai et al., 2020a; Tosh et al., 2020) suggests the contrastively learned representations can lead to a good downstream performance. Beyond the theory, Tian et al. (2020) shows what matters more for the performance are 1) the choice of the contrastive learning objective; and 2) the creation of the positive and negative data pairs in the contrastive objective. Recent work (Khosla et al., 2020) extends the usage of contrastive learning from the self-supervised setting to the supervised setting. The supervised setting defines the positive pairs as the data from the same class in the contrastive objective, while the self-supervised setting defines the positive pairs as the data with different augmentations.

Our work also closely rates to the *skewed divergence* measurement between distributions (Lee, 1999; 2001; Nielsen, 2010; Yamada et al., 2013). Recall that the usage of the relative parameters plays a crucial role to regularize our objective for its boundness and low variance. This idea is similar to the *skewed divergence* measurement, that when calculating the divergence between distributions $P$ and $Q$, instead of considering $\mathrm{D}(P \,\|\, Q)$, these approaches consider $\mathrm{D}(P \,\|\, \alpha P + (1 - \alpha)Q)$ with $D$ representing the divergence and $0 < \alpha < 1$. A natural example is that the Jensen-Shannon divergence is a symmetric skewed KL divergence: $D_{\mathrm{JS}}(P \,\|\, Q) = 0.5D_{\mathrm{KL}}(P \,\|\, 0.5P + 0.5Q) + 0.5D_{\mathrm{KL}}(Q \,\|\, 0.5P + 0.5Q)$. Compared to the non-skewed counterpart, the skewed divergence has shown to have a more robust estimation for its value (Lee, 1999; 2001; Yamada et al., 2013). Different from these works that focus on estimating the values of distribution divergence, we focus on learning self-supervised representations.

## 5 CONCLUSION

In this work, we present RPC, the Relative Predictive Coding, that achieves a good balance among the three challenges when modeling a contrastive learning objective: training stability, sensitivity to minibatch size, and downstream task performance. We believe this work brings an appealing option for training self-supervised models and inspires future work to design objectives for balancing the aforementioned three challenges. In the future, we are interested in applying RPC in other application domains and developing more principled approaches for better representation learning.

## ACKNOWLEDGEMENT

This work was supported in part by the NSF IIS1763562, NSF Awards #1750439 #1722822, National Institutes of Health, IARPA D17PC00340, ONR Grant N000141812861, and Facebook PhD Fellowship. We would also like to acknowledge NVIDIA's GPU support and Cloud TPU support from Google's TensorFlow Research Cloud (TFRC).

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

# A  APPENDIX

## A.1  PROOF OF LEMMA 1 IN THE MAIN TEXT

**Lemma 2 (Optimal Solution for $J_{\mathrm{RPC}}$, restating Lemma 1 in the main text)** *Let*

$$J_{\mathrm{RPC}}(X,Y) := \sup_{f \in \mathcal{F}} \mathbb{E}_{P_{XY}}[f(x,y)] - \alpha \mathbb{E}_{P_X P_Y}[f(x,y)] - \frac{\beta}{2} \mathbb{E}_{P_{XY}}\left[f^2(x,y)\right] - \frac{\gamma}{2} \mathbb{E}_{P_X P_Y}\left[f^2(x,y)\right]$$

*and $r(x,y) = \frac{p(x,y)}{p(x)p(y)}$ be the density ratio. $J_{\mathrm{RPC}}$ has the optimal solution*

$$f^*(x,y) = \frac{r(x,y) - \alpha}{\beta\, r(x,y) + \gamma} := r_{\alpha,\beta,\gamma}(x,y) \text{ with } -\frac{\alpha}{\gamma} \le r_{\alpha,\beta,\gamma} \le \frac{1}{\beta}.$$

*Proof:* The second-order functional derivative of the objective is

$$-\beta dP_{X,Y} - \gamma dP_X P_Y,$$

which is always negative. The negative second-order functional derivative implies the objective has a supreme value. Then, take the first-order functional derivative $\frac{\partial J_{\mathrm{RPC}}}{\partial m}$ and set it to zero:

$$dP_{X,Y} - \alpha \cdot dP_X P_Y - \beta \cdot f(x,y) \cdot dP_{X,Y} - \gamma \cdot f(x,y) \cdot dP_X P_Y = 0.$$

We then get

$$f^*(x,y) = \frac{dP_{X,Y} - \alpha \cdot dP_X P_Y}{\beta \cdot dP_{X,Y} + \gamma \cdot dP_X P_Y} = \frac{p(x,y) - \alpha p(x)p(y)}{\beta p(x,y) + \gamma p(x)p(y)} = \frac{r(x,y) - \alpha}{\beta r(x,y) + \gamma}.$$

Since $0 \le r(x,y) \le \infty$, we have $-\frac{\alpha}{\gamma} \le \frac{r(x,y)-\alpha}{\beta r(x,y)+\gamma} \le \frac{1}{\beta}$. Hence,

$$\forall \beta \ne 0, \gamma \ne 0, f^*(x,y) := r_{\alpha,\beta,\gamma}(x,y) \text{ with } -\frac{\alpha}{\gamma} \le r_{\alpha,\beta,\gamma} \le \frac{1}{\beta}.$$

$\square$

## A.2  RELATION BETWEEN $J_{\mathrm{RPC}}$ AND $D_{\chi^2}$

In this subsection, we aim to show the following: 1) $D_{\chi^2}(P_{XY} \,\|\, P_X P_Y) = \mathbb{E}_{P_X P_Y}[r^2(x,y)] - 1$; and 2) $J_{\mathrm{RPC}}(X,Y) = \frac{\beta+\gamma}{2}\mathbb{E}_{P'}[r^2_{\alpha,\beta,\gamma}(x,y)]$ by having $P' = \frac{\beta}{\beta+\gamma}P_{XY} + \frac{\gamma}{\beta+\gamma}P_X P_Y$ as the mixture distribution of $P_{XY}$ and $P_X P_Y$.

**Lemma 3**  $D_{\chi^2}(P_{XY} \,\|\, P_X P_Y) = \mathbb{E}_{P_X P_Y}[r^2(x,y)] - 1$

*Proof:* By definition (Nielsen & Nock, 2013),

$$\begin{aligned}
D_{\chi^2}(P_{XY} \,\|\, P_X P_Y) &= \int \frac{\left(dP_{XY}\right)^2}{dP_X P_Y} - 1 = \int \left(\frac{dP_{XY}}{dP_X P_Y}\right)^2 dP_X P_Y - 1 \\
&= \int \left(\frac{p(x,y)}{p(x)p(y)}\right)^2 dP_X P_Y - 1 = \int r^2(x,y) dP_X P_Y - 1 \\
&= \mathbb{E}_{P_X P_Y}[r^2(x,y)] - 1.
\end{aligned}$$

$\square$

**Lemma 4** *Defining $P' = \frac{\beta}{\beta+\gamma}P_{XY} + \frac{\gamma}{\beta+\gamma}P_X P_Y$ as a mixture distribution of $P_{XY}$ and $P_X P_Y$, $J_{\mathrm{RPC}}(X,Y) = \frac{\beta+\gamma}{2}\mathbb{E}_{P'}[r^2_{\alpha,\beta,\gamma}(x,y)]$.*

*Proof:* Plug in the optimal solution $f^*(x, y) = \frac{dP_{X,Y} - \alpha \cdot dP_X P_Y}{\beta \cdot dP_{X,Y} + \gamma \cdot dP_X P_Y}$ (see Lemma 2) into $J_{\text{RPC}}$:

$$
\begin{aligned}
J_{\text{RPC}} &= \mathbb{E}_{P_{XY}}[f^*(x, y)] - \alpha \mathbb{E}_{P_X P_Y}[f^*(x, y)] - \frac{\beta}{2} \mathbb{E}_{P_{XY}}\left[f^{*2}(x, y)\right] - \frac{\gamma}{2} \mathbb{E}_{P_X P_Y}\left[f^{*2}(x, y)\right] \\
&= \int f^*(x, y) \cdot \left(dP_{XY} - \alpha \cdot dP_X P_Y\right) - \frac{1}{2} f^{*2}(x, y) \cdot \left(\beta \cdot dP_{XY} + \gamma \cdot dP_X P_Y\right) \\
&= \int \frac{dP_{X,Y} - \alpha \cdot dP_X P_Y}{\beta \cdot dP_{X,Y} + \gamma \cdot dP_X P_Y}\left(dP_{XY} - \alpha \cdot dP_X P_Y\right) - \frac{1}{2}\left(\frac{dP_{X,Y} - \alpha \cdot dP_X P_Y}{\beta \cdot dP_{X,Y} + \gamma \cdot dP_X P_Y}\right)^2 \left(\beta \cdot dP_{XY} + \gamma \cdot dP_X P_Y\right) \\
&= \frac{1}{2} \int \left(\frac{dP_{X,Y} - \alpha \cdot dP_X P_Y}{\beta \cdot dP_{X,Y} + \gamma \cdot dP_X P_Y}\right)^2 \left(\beta \cdot dP_{XY} + \gamma \cdot dP_X P_Y\right) \\
&= \frac{\beta + \gamma}{2} \int \left(\frac{dP_{X,Y} - \alpha \cdot dP_X P_Y}{\beta \cdot dP_{X,Y} + \gamma \cdot dP_X P_Y}\right)^2 \left(\frac{\beta}{\beta + \gamma} \cdot dP_{XY} + \frac{\gamma}{\beta + \gamma} \cdot dP_X P_Y\right).
\end{aligned}
$$

Since we define $r_{\alpha,\beta,\gamma} = \frac{dP_{X,Y} - \alpha \cdot dP_X P_Y}{\beta \cdot dP_{X,Y} + \gamma \cdot dP_X P_Y}$ and $P' = \frac{\beta}{\beta + \gamma} P_{XY} + \frac{\gamma}{\beta + \gamma} P_X P_Y$,

$$
J_{\text{RPC}} = \frac{\beta + \gamma}{2} \mathbb{E}_{P'}[r^2_{\alpha,\beta,\gamma}(x, y)].
$$

$\square$

## A.3 PROOF OF PROPOSITION 1 IN THE MAIN TEXT

The proof contains two parts: showing $0 \leq J_{\text{RPC}} \leq \frac{1}{2\beta} + \frac{\alpha^2}{2\gamma}$ (see Section A.3.1) and $\hat{J}^{m,n}_{\text{RPC}}$ is a consistent estimator for $J_{\text{RPC}}$ (see Section A.3.2).

### A.3.1 BOUNDNESS OF $J_{\text{RPC}}$

**Lemma 5 (Boundness of $J_{\text{RPC}}$)** $0 \leq J_{\text{RPC}} \leq \frac{1}{2\beta} + \frac{\alpha^2}{2\gamma}$

*Proof:* Lemma 4 suggests $J_{\text{RPC}}(X, Y) = \frac{\beta + \gamma}{2} \mathbb{E}_{P'}[r^2_{\alpha,\beta,\gamma}(x, y)]$ with $P' = \frac{\beta}{\beta + \gamma} P_{XY} + \frac{\gamma}{\beta + \gamma} P_X P_Y$ as the mixture distribution of $P_{XY}$ and $P_X P_Y$. Hence, it is obvious $J_{\text{RPC}}(X, Y) \geq 0$.

We leverage the intermediate results in the proof of Lemma 4:

$$
\begin{aligned}
J_{\text{RPC}}(X, Y) &= \frac{1}{2} \int \left(\frac{dP_{X,Y} - \alpha \cdot dP_X P_Y}{\beta \cdot dP_{X,Y} + \gamma \cdot dP_X P_Y}\right)^2 \left(\beta \cdot dP_{XY} + \gamma \cdot dP_X P_Y\right) \\
&= \frac{1}{2} \int dP_{X,Y}\left(\frac{dP_{X,Y} - \alpha \cdot dP_X P_Y}{\beta \cdot dP_{X,Y} + \gamma \cdot dP_X P_Y}\right) - \frac{\alpha}{2} \int dP_X P_Y\left(\frac{dP_{X,Y} - \alpha \cdot dP_X P_Y}{\beta \cdot dP_{X,Y} + \gamma \cdot dP_X P_Y}\right) \\
&= \frac{1}{2} \mathbb{E}_{P_{XY}}[r_{\alpha,\beta,\gamma}(x, y)] - \frac{\alpha}{2} \mathbb{E}_{P_X P_Y}[r_{\alpha,\beta,\gamma}(x, y)].
\end{aligned}
$$

Since $-\frac{\alpha}{\gamma} \leq r_{\alpha,\beta,\gamma} \leq \frac{1}{\beta}$, $J_{\text{RPC}}(X, Y) \leq \frac{1}{2\beta} + \frac{\alpha^2}{2\gamma}$. $\square$

### A.3.2 CONSISTENCY

We first recall the definition of the estimation of $J_{\text{RPC}}$:

**Definition 2 ($\hat{J}^{m,n}_{\text{RPC}}$, empirical estimation of $J_{\text{RPC}}$, restating Definition 1 in the main text)** *We parametrize $f$ via a family of neural networks $\mathcal{F}_\Theta := \{f_\theta : \theta \in \Theta \subseteq \mathbb{R}^d\}$ where $d \in \mathbb{N}$ and $\Theta$ is compact. Let $\{x_i, y_i\}_{i=1}^n$ be $n$ samples drawn uniformly at random from $P_{XY}$ and $\{x'_j, y'_j\}_{j=1}^m$ be $m$ samples drawn uniformly at random from $P_X P_Y$. Then,*

$$
\hat{J}^{m,n}_{\text{RPC}} = \sup_{f_\theta \in \mathcal{F}_\Theta} \frac{1}{n} \sum_{i=1}^n f_\theta(x_i, y_i) - \frac{1}{m} \sum_{j=1}^m \alpha f_\theta(x'_j, y'_j) - \frac{1}{n} \sum_{i=1}^n \frac{\beta}{2} f_\theta^2(x_i, y_i) - \frac{1}{m} \sum_{j=1}^m \frac{\gamma}{2} f_\theta^2(x'_j, y'_j).
$$

Our goal is to show that $\hat{J}_{\mathrm{RPC}}^{m,n}$ is a consistent estimator for $J_{\mathrm{RPC}}$. We begin with the following definition:

$$\hat{J}_{\mathrm{RPC},\theta}^{m,n} := \frac{1}{n}\sum_{i=1}^{n} f_\theta(x_i, y_i) - \frac{1}{m}\sum_{j=1}^{m} \alpha f_\theta(x_j', y_j') - \frac{1}{n}\sum_{i=1}^{n} \frac{\beta}{2} f_\theta^2(x_i, y_i) - \frac{1}{m}\sum_{j=1}^{m} \frac{\gamma}{2} f_\theta^2(x_j', y_j') \quad (3)$$

and

$$\mathbb{E}\left[\hat{J}_{\mathrm{RPC},\theta}\right] := \mathbb{E}_{P_{XY}}[f_\theta(x,y)] - \alpha\mathbb{E}_{P_X P_Y}[f_\theta(x,y)] - \frac{\beta}{2}\mathbb{E}_{P_{XY}}[f_\theta^2(x,y)] - \frac{\gamma}{2}\mathbb{E}_{P_X P_Y}[f_\theta^2(x,y)]. \quad (4)$$

Then, we follow the steps:

- The first part is about estimation. We show that, with high probability, $\hat{J}_{\mathrm{RPC},\theta}^{m,n}$ is close to $\mathbb{E}\left[\hat{J}_{\mathrm{RPC},\theta}\right]$, for any given $\theta$.

- The second part is about approximation. We will apply the universal approximation lemma of neural networks (Hornik et al., 1989) to show that there exists a network $\theta^*$ such that $\mathbb{E}\left[\hat{J}_{\mathrm{RPC},\theta^*}\right]$ is close to $J_{\mathrm{RPC}}$.

**Part I - Estimation: With high probability, $\hat{J}_{\mathrm{RPC},\theta}^{m,n}$ is close to $\mathbb{E}\left[\hat{J}_{\mathrm{RPC},\theta}\right]$, for any given $\theta$.**
Throughout the analysis on the uniform convergence, we need the assumptions on the boundness and smoothness of the function $f_\theta$. Since we show the optimal function $f$ is bounded in $J_{\mathrm{RPC}}$, we can use the same bounded values for $f_\theta$ without losing too much precision. The smoothness of the function suggests that the output of the network should only change slightly when only slightly perturbing the parameters. Specifically, the two assumptions are as follows:

**Assumption 1 (boundness of $f_\theta$)** *There exist universal constants such that $\forall f_\theta \in \mathcal{F}_\Theta$, $C_L \leq f_\theta \leq C_U$. For notations simplicity, we let $M = C_U - C_L$ be the range of $f_\theta$ and $U = \max\{|C_U|, |C_L|\}$ be the maximal absolute value of $f_\theta$. In the paper, we can choose to constrain that $C_L = -\frac{\alpha}{\gamma}$ and $C_U = \frac{1}{\beta}$ since the optimal function $f^*$ has $-\frac{\alpha}{\gamma} \leq f^* \leq \frac{1}{\beta}$.*

**Assumption 2 (smoothness of $f_\theta$)** *There exists constant $\rho > 0$ such that $\forall(x,y) \in (\mathcal{X} \times \mathcal{Y})$ and $\theta_1, \theta_2 \in \Theta$, $|f_{\theta_1}(x,y) - f_{\theta_2}(x,y)| \leq \rho|\theta_1 - \theta_2|$.*

Now, we can bound the rate of uniform convergence of a function class in terms of covering number (Bartlett, 1998):

**Lemma 6 (Estimation)** *Let $\epsilon > 0$ and $\mathcal{N}(\Theta, \epsilon)$ be the covering number of $\Theta$ with radius $\epsilon$. Then,*

$$\Pr\left(\sup_{f_\theta \in \mathcal{F}_\Theta} \left|\hat{J}_{\mathrm{RPC},\theta}^{m,n} - \mathbb{E}\left[\hat{J}_{\mathrm{RPC},\theta}\right]\right| \geq \epsilon\right)$$

$$\leq 2\mathcal{N}(\Theta, \frac{\epsilon}{4\rho\left(1 + \alpha + 2(\beta + \gamma)U\right)})\left(\exp\left(-\frac{n\epsilon^2}{32M^2}\right) + \exp\left(-\frac{m\epsilon^2}{32M^2\alpha^2}\right) + \exp\left(-\frac{n\epsilon^2}{32U^2\beta^2}\right) + \exp\left(-\frac{m\epsilon^2}{32U^2\gamma^2}\right)\right).$$

*Proof:* For notation simplicity, we define the operators

- $P(f) = \mathbb{E}_{P_{XY}}[f(x,y)]$ and $P_n(f) = \frac{1}{n}\sum_{i=1}^{n} f(x_i, y_i)$
- $Q(f) = \mathbb{E}_{P_X P_Y}[f(x,y)]$ and $Q_m(f) = \frac{1}{m}\sum_{j=1}^{m} f(x_j', y_j')$

Hence,

$$\left|\hat{J}_{\mathrm{RPC},\theta}^{m,n} - \mathbb{E}\left[\hat{J}_{\mathrm{RPC},\theta}\right]\right|$$

$$= \left|P_n(f_\theta) - P(f_\theta) - \alpha Q_m(f_\theta) + \alpha Q(f_\theta) - \beta P_n(f_\theta^2) + \beta P(f_\theta^2) - \gamma Q_m(f_\theta^2) + \gamma Q(f_\theta^2)\right|$$

$$\leq |P_n(f_\theta) - P(f_\theta)| + \alpha|Q_m(f_\theta) - Q(f_\theta)| + \beta\left|P_n(f_\theta^2) - P(f_\theta^2)\right| + \gamma\left|Q_m(f_\theta^2) - Q(f_\theta^2)\right|$$

Let $\epsilon' = \frac{\epsilon}{4\rho\left(1+\alpha+2(\beta+\gamma)U\right)}$ and $T := \mathcal{N}(\Theta, \epsilon')$. Let $C = \{f_{\theta_1}, f_{\theta_2}, \cdots, f_{\theta_T}\}$ with $\{\theta_1, \theta_2, \cdots, \theta_T\}$ be such that $B_\infty(\theta_1, \epsilon'), \cdots, B_\infty(\theta_T, \epsilon')$ are $\epsilon'$ cover. Hence, for any $f_\theta \in \mathcal{F}_\Theta$, there is an $f_{\theta_k} \in C$ such that $\|\theta - \theta_k\|_\infty \le \epsilon'$.

Then, for any $f_{\theta_k} \in C$:

$$\left|\hat{J}_{\mathrm{RPC},\theta}^{m,n} - \mathbb{E}\left[\hat{J}_{\mathrm{RPC},\theta}\right]\right|$$

$$\le |P_n(f_\theta) - P(f_\theta)| + \alpha\,|Q_m(f_\theta) - Q(f_\theta)| + \beta\left|P_n(f_\theta^2) - P(f_\theta^2)\right| + \gamma\left|Q_m(f_\theta^2) - Q(f_\theta^2)\right|$$

$$\le |P_n(f_{\theta_k}) - P(f_{\theta_k})| + |P_n(f_\theta) - P_n(f_{\theta_k})| + |P(f_\theta) - P(f_{\theta_k})|$$

$$+ \alpha\bigg(\,|Q_m(f_{\theta_k}) - Q(f_{\theta_k})| + |Q_m(f_\theta) - Q_m(f_{\theta_k})| + |Q(f_\theta) - Q(f_{\theta_k})|\,\bigg)$$

$$+ \beta\bigg(\left|P_n(f_{\theta_k}^2) - P(f_{\theta_k}^2)\right| + \left|P_n(f_\theta^2) - P_n(f_{\theta_k}^2)\right| + \left|P(f_\theta^2) - P(f_{\theta_k}^2)\right|\bigg)$$

$$+ \gamma\bigg(\left|Q_m(f_{\theta_k}^2) - Q(f_{\theta_k}^2)\right| + \left|Q_m(f_\theta^2) - Q_m(f_{\theta_k}^2)\right| + \left|Q(f_\theta^2) - Q(f_{\theta_k}^2)\right|\bigg)$$

$$\le |P_n(f_{\theta_k}) - P(f_{\theta_k})| + \rho\|\theta - \theta_k\| + \rho\|\theta - \theta_k\|$$

$$+ \alpha\bigg(\,|Q_m(f_{\theta_k}) - Q(f_{\theta_k})| + \rho\|\theta - \theta_k\| + \rho\|\theta - \theta_k\|\,\bigg)$$

$$+ \beta\bigg(\left|P_n(f_{\theta_k}^2) - P(f_{\theta_k}^2)\right| + 2\rho U\|\theta - \theta_k\| + 2\rho U\|\theta - \theta_k\|\bigg)$$

$$+ \gamma\bigg(\left|Q_m(f_{\theta_k}^2) - Q(f_{\theta_k}^2)\right| + 2\rho U\|\theta - \theta_k\| + 2\rho U\|\theta - \theta_k\|\bigg)$$

$$= |P_n(f_{\theta_k}) - P(f_{\theta_k})| + \alpha\,|Q_m(f_{\theta_k}) - Q(f_{\theta_k})| + \beta\left|P_n(f_{\theta_k}^2) - P(f_{\theta_k}^2)\right| + \gamma\left|Q_m(f_{\theta_k}^2) - Q(f_{\theta_k}^2)\right|$$

$$+ 2\rho\left(1 + \alpha + 2(\beta+\gamma)U\right)\|\theta - \theta_k\|$$

$$\le |P_n(f_{\theta_k}) - P(f_{\theta_k})| + \alpha\,|Q_m(f_{\theta_k}) - Q(f_{\theta_k})| + \beta\left|P_n(f_{\theta_k}^2) - P(f_{\theta_k}^2)\right| + \gamma\left|Q_m(f_{\theta_k}^2) - Q(f_{\theta_k}^2)\right| + \frac{\epsilon}{2},$$

where

- $|P_n(f_\theta) - P_n(f_{\theta_k})| \le \rho\|\theta - \theta_k\|$ due to Assumption 2, and the result also applies for $|P(f_\theta) - P(f_{\theta_k})|$, $|Q_m(f_\theta) - Q_m(f_{\theta_k})|$, and $|Q(f_\theta) - Q(f_{\theta_k})|$.
- $\left|P_n(f_\theta^2) - P_n(f_{\theta_k}^2)\right| \le 2\|f_\theta\|_\infty \rho\|\theta - \theta_k\| \le 2\rho U\|\theta - \theta_k\|$ due to Assumptions 1 and 2. The result also applies for $\left|P(f_\theta^2) - P(f_{\theta_k}^2)\right|$, $\left|Q_m(f_\theta^2) - Q_m(f_{\theta_k}^2)\right|$, and $\left|Q(f_\theta^2) - Q(f_{\theta_k}^2)\right|$.

Hence,

$$\Pr\left(\sup_{f_\theta \in \mathcal{F}_\Theta} \left|\hat{J}_{\mathrm{RPC},\theta}^{m,n} - \mathbb{E}\left[\hat{J}_{\mathrm{RPC},\theta}\right]\right| \ge \epsilon\right)$$

$$\le \Pr\left(\max_{f_{\theta_k} \in C} |P_n(f_{\theta_k}) - P(f_{\theta_k})| + \alpha\,|Q_m(f_{\theta_k}) - Q(f_{\theta_k})| + \beta\left|P_n(f_{\theta_k}^2) - P(f_{\theta_k}^2)\right| + \gamma\left|Q_m(f_{\theta_k}^2) - Q(f_{\theta_k}^2)\right| + \frac{\epsilon}{2} \ge \epsilon\right)$$

$$= \Pr\left(\max_{f_{\theta_k} \in C} |P_n(f_{\theta_k}) - P(f_{\theta_k})| + \alpha\,|Q_m(f_{\theta_k}) - Q(f_{\theta_k})| + \beta\left|P_n(f_{\theta_k}^2) - P(f_{\theta_k}^2)\right| + \gamma\left|Q_m(f_{\theta_k}^2) - Q(f_{\theta_k}^2)\right| \ge \frac{\epsilon}{2}\right)$$

$$\le \sum_{k=1}^T \Pr\left(|P_n(f_{\theta_k}) - P(f_{\theta_k})| + \alpha\,|Q_m(f_{\theta_k}) - Q(f_{\theta_k})| + \beta\left|P_n(f_{\theta_k}^2) - P(f_{\theta_k}^2)\right| + \gamma\left|Q_m(f_{\theta_k}^2) - Q(f_{\theta_k}^2)\right| \ge \frac{\epsilon}{2}\right)$$

$$\le \sum_{k=1}^T \Pr\left(|P_n(f_{\theta_k}) - P(f_{\theta_k})| \ge \frac{\epsilon}{8}\right) + \Pr\left(\alpha\,|Q_m(f_{\theta_k}) - Q(f_{\theta_k})| \ge \frac{\epsilon}{8}\right)$$

$$+ \Pr\left(\beta\left|P_n(f_{\theta_k}^2) - P(f_{\theta_k}^2)\right| \ge \frac{\epsilon}{8}\right) + \Pr\left(\gamma\left|Q_m(f_{\theta_k}^2) - Q(f_{\theta_k}^2)\right| \ge \frac{\epsilon}{8}\right).$$

With Hoeffding's inequality,

- $\Pr\left(|P_n(f_{\theta_k}) - P(f_{\theta_k})| \geq \frac{\epsilon}{8}\right) \leq 2\exp\left(-\frac{n\epsilon^2}{32M^2}\right)$

- $\Pr\left(\alpha\left|Q_m(f_{\theta_k}) - Q(f_{\theta_k})\right| \geq \frac{\epsilon}{8}\right) \leq 2\exp\left(-\frac{m\epsilon^2}{32M^2\alpha^2}\right)$

- $\Pr\left(\beta\left|P_n(f_{\theta_k}^2) - P(f_{\theta_k}^2)\right| \geq \frac{\epsilon}{8}\right) \leq 2\exp\left(-\frac{n\epsilon^2}{32U^2\beta^2}\right)$

- $\Pr\left(\gamma\left|Q_m(f_{\theta_k}^2) - Q(f_{\theta_k}^2)\right| \geq \frac{\epsilon}{8}\right) \leq 2\exp\left(-\frac{m\epsilon^2}{32U^2\gamma^2}\right)$

To conclude,

$$\Pr\left(\sup_{f_\theta \in \mathcal{F}_\Theta} \left|\hat{J}_{\mathrm{RPC},\theta}^{m,n} - \mathbb{E}\left[\hat{J}_{\mathrm{RPC},\theta}\right]\right| \geq \epsilon\right)$$

$$\leq 2\mathcal{N}(\Theta, \frac{\epsilon}{4\rho\left(1+\alpha+2(\beta+\gamma)U\right)})\left(\exp\left(-\frac{n\epsilon^2}{32M^2}\right) + \exp\left(-\frac{m\epsilon^2}{32M^2\alpha^2}\right) + \exp\left(-\frac{n\epsilon^2}{32U^2\beta^2}\right) + \exp\left(-\frac{m\epsilon^2}{32U^2\gamma^2}\right)\right).$$

$\square$

**Part II - Approximation: Neural Network Universal Approximation.** We leverage the universal function approximation lemma of neural network

**Lemma 7 (Approximation (Hornik et al., 1989))** *Let $\epsilon > 0$. There exists $d \in \mathbb{N}$ and a family of neural networks $\mathcal{F}_\Theta := \{f_\theta : \theta \in \Theta \subseteq \mathbb{R}^d\}$ where $\Theta$ is compact, such that* $\inf_{f_\theta \in \mathcal{F}_\Theta} \left|\mathbb{E}\left[\hat{J}_{\mathrm{RPC},\theta}\right] - J_{\mathrm{RPC}}\right| \leq \epsilon.$

**Part III - Bringing everything together.** Now, we are ready to bring the estimation and approximation together to show that there exists a neural network $\theta^*$ such that, with high probability, $\hat{J}_{\mathrm{RPC},\theta}^{m,n}$ can approximate $J_{\mathrm{RPC}}$ with $n' = \min\{n, m\}$ at a rate of $O(1/\sqrt{n'})$:

**Proposition 3** *With probability at least $1 - \delta$, $\exists \theta^* \in \Theta$, $|J_{\mathrm{RPC}} - \hat{J}_{\mathrm{RPC},\theta}^{m,n}| = O(\sqrt{\frac{d+\log(1/\delta)}{n'}})$, where $n' = \min\{n, m\}$.*

*Proof:* The proof follows by combining Lemma 6 and 7.

First, Lemma 7 suggests, $\exists \theta^* \in \Theta$,

$$\left|\mathbb{E}\left[\hat{J}_{\mathrm{RPC},\theta^*}\right] - J_{\mathrm{RPC}}\right| \leq \frac{\epsilon}{2}.$$

Next, we perform analysis on the estimation error, aiming to find $n, m$ and the corresponding probability, such that

$$\left|\hat{J}_{\mathrm{RPC},\theta}^{m,n} - \mathbb{E}\left[\hat{J}_{\mathrm{RPC},\theta^*}\right]\right| \leq \frac{\epsilon}{2}.$$

Applying Lemma 6 with the covering number of the neural network: $\Big(\mathcal{N}(\Theta, \epsilon) = O\Big(\exp\big(d\log(1/\epsilon)\big)\Big)$ (Anthony & Bartlett, 2009)$\Big)$ and let $n' = \min\{n, m\}$:

$$\Pr\left(\sup_{f_\theta \in \mathcal{F}_\Theta} \left|\hat{J}_{\mathrm{RPC},\theta}^{m,n} - \mathbb{E}\left[\hat{J}_{\mathrm{RPC},\theta}\right]\right| \geq \frac{\epsilon}{2}\right)$$

$$\leq 2\mathcal{N}(\Theta, \frac{\epsilon}{8\rho\left(1+\alpha+2(\beta+\gamma)U\right)})\left(\exp\left(-\frac{n\epsilon^2}{128M^2}\right) + \exp\left(-\frac{m\epsilon^2}{128M^2\alpha^2}\right) + \exp\left(-\frac{n\epsilon^2}{128U^2\beta^2}\right) + \exp\left(-\frac{m\epsilon^2}{128U^2\gamma^2}\right)\right)$$

$$= O\Big(\exp\big(d\log(1/\epsilon) - n'\epsilon^2\big)\Big),$$

where the big-O notation absorbs all the constants that do not require in the following derivation. Since we want to bound the probability with $1 - \delta$, we solve the $\epsilon$ such that

$$\exp\big(d\log\left(1/\epsilon\right) - n'\epsilon^2\big) \leq \delta.$$

With $\log\left(x\right) \leq x - 1$,

$$n'\epsilon^2 + d(\epsilon - 1) \geq n'\epsilon^2 + d\log\epsilon \geq \log\left(1/\delta\right),$$

where this inequality holds when

$$\epsilon = O\bigg(\sqrt{\frac{d + \log\left(1/\delta\right)}{n'}}\bigg).$$

$\square$

## A.4 Proof of Proposition 2 in the Main Text - From an Asymptotic Viewpoint

Here, we provide the variance analysis on $\hat{J}_{\text{RPC}}^{m,n}$ via an asymptotic viewpoint. First, assuming the network is correctly specified, and hence there exists a network parameter $\theta^*$ satisfying $f^*(x,y) = f_{\theta^*}(x,y) = r_{\alpha,\beta,\gamma}(x,y)$. Then we recall that $\hat{J}_{\text{RPC}}^{m,n}$ is a consistent estimator of $J^{\text{RPC}}$ (see Proposition 3), and under regular conditions, the estimated network parameter $\hat{\theta}$ in $\hat{J}_{\text{RPC}}^{m,n}$ satisfying the asymptotic normality in the large sample limit (see Theorem 5.23 in (Van der Vaart, 2000)). We recall the definition of $\hat{J}_{\text{RPC},\theta}^{m,n}$ in equation 3 and let $n' = \min\{n,m\}$, the asymptotic expansion of $\hat{J}_{\text{RPC}}^{m,n}$ has

$$\begin{aligned}
\hat{J}_{\text{RPC},\theta^*}^{m,n} &= \hat{J}_{\text{RPC},\hat{\theta}}^{m,n} + \dot{\hat{J}}_{\text{RPC},\hat{\theta}}^{m,n}(\theta^* - \hat{\theta}) + o(\|\theta^* - \hat{\theta}\|) \\
&= \hat{J}_{\text{RPC},\hat{\theta}}^{m,n} + \dot{\hat{J}}_{\text{RPC},\hat{\theta}}^{m,n}(\theta^* - \hat{\theta}) + o_p(\frac{1}{\sqrt{n'}}) \\
&= \hat{J}_{\text{RPC},\hat{\theta}}^{m,n} + o_p(\frac{1}{\sqrt{n'}}),
\end{aligned} \tag{5}$$

where $\dot{\hat{J}}_{\text{RPC},\hat{\theta}}^{m,n} = 0$ since $\hat{\theta}$ is the estimation from $\hat{J}_{\text{RPC}}^{m,n} = \sup\limits_{f_\theta \in \mathcal{F}_\Theta} \hat{J}_{\text{RPC},\theta}^{m,n}$.

Next, we recall the definition in equation 4:

$$\mathbb{E}[\hat{J}_{\text{RPC},\hat{\theta}}] = \mathbb{E}_{P_{XY}}[f_{\hat{\theta}}(x,y)] - \alpha\mathbb{E}_{P_X P_Y}[f_{\hat{\theta}}(x,y)] - \frac{\beta}{2}\mathbb{E}_{P_{XY}}[f_{\hat{\theta}}^2(x,y)] - \frac{\gamma}{2}\mathbb{E}_{P_X P_Y}[f_{\hat{\theta}}^2(x,y)].$$

Likewise, the asymptotic expansion of $\mathbb{E}[\hat{J}_{\text{RPC},\theta}]$ has

$$\begin{aligned}
\mathbb{E}[\hat{J}_{\text{RPC},\hat{\theta}}] &= \mathbb{E}[\hat{J}_{\text{RPC},\theta^*}] + \mathbb{E}[\dot{\hat{J}}_{\text{RPC},\theta^*}](\hat{\theta} - \theta^*) + o(\|\hat{\theta} - \theta^*\|) \\
&= \mathbb{E}[\hat{J}_{\text{RPC},\theta^*}] + \mathbb{E}[\dot{\hat{J}}_{\text{RPC},\theta^*}](\hat{\theta} - \theta^*) + o_p(\frac{1}{\sqrt{n'}}) \\
&= \mathbb{E}[\hat{J}_{\text{RPC},\theta^*}] + o_p(\frac{1}{\sqrt{n'}}),
\end{aligned} \tag{6}$$

where $\mathbb{E}[\dot{\hat{J}}_{\text{RPC},\theta^*}] = 0$ since $\mathbb{E}[\hat{J}_{\text{RPC},\theta^*}] = J_{\text{RPC}}$ and $\theta^*$ satisfying $f^*(x,y) = f_{\theta^*}(x,y)$.

Combining equations 5 and 6:

$$
\begin{aligned}
\hat{J}^{m,n}_{\mathrm{RPC},\hat{\theta}} - \mathbb{E}[\hat{J}_{\mathrm{RPC},\hat{\theta}}] =& \hat{J}^{m,n}_{\mathrm{RPC},\theta^*} - J_{\mathrm{RPC}} + o_p(\frac{1}{\sqrt{n'}}) \\
=& \frac{1}{n}\sum_{i=1}^{n} f^*_\theta(x_i,y_i) - \alpha\frac{1}{m}\sum_{j=1}^{m} f^*_\theta(x'_j,y'_j) - \frac{\beta}{2}\frac{1}{n}\sum_{i=1}^{n} f^2_{\theta^*}(x_i,y_i) - \frac{\gamma}{2}\frac{1}{m}\sum_{j=1}^{m} f^2_{\theta^*}(x'_j,y'_j) \\
& - \mathbb{E}_{P_{XY}}[f^*(x,y)] + \alpha\mathbb{E}_{P_X P_Y}[f^*(x,y)] + \frac{\beta}{2}\mathbb{E}_{P_{XY}}\left[f^{*2}(x,y)\right] + \frac{\gamma}{2}\mathbb{E}_{P_X P_Y}\left[f^{*2}(x,y)\right] + o_p(\frac{1}{\sqrt{n'}}) \\
=& \frac{1}{n}\sum_{i=1}^{n} r_{\alpha,\beta,\gamma}(x_i,y_i) - \alpha\frac{1}{m}\sum_{j=1}^{m} r_{\alpha,\beta,\gamma}(x'_j,y'_j) - \frac{\beta}{2}\frac{1}{n}\sum_{i=1}^{n} r^2_{\alpha,\beta,\gamma}(x_i,y_i) - \frac{\gamma}{2}\frac{1}{m}\sum_{j=1}^{m} r^2_{\alpha,\beta,\gamma}(x'_j,y'_j) \\
& - \mathbb{E}_{P_{XY}}[r_{\alpha,\beta,\gamma}(x,y)] + \alpha\mathbb{E}_{P_X P_Y}[r_{\alpha,\beta,\gamma}(x,y)] + \frac{\beta}{2}\mathbb{E}_{P_{XY}}\left[r^2_{\alpha,\beta,\gamma}(x,y)\right] + \frac{\gamma}{2}\mathbb{E}_{P_X P_Y}\left[r^2_{\alpha,\beta,\gamma}(x,y)\right] \\
& + o_p(\frac{1}{\sqrt{n'}}) \\
=& \frac{1}{\sqrt{n}}\cdot\frac{1}{\sqrt{n}}\sum_{i=1}^{n}\left(r_{\alpha,\beta,\gamma}(x_i,y_i) - \frac{\beta}{2}r^2_{\alpha,\beta,\gamma}(x_i,y_i) - \mathbb{E}_{P_{XY}}\left[r_{\alpha,\beta,\gamma}(x,y) - \frac{\beta}{2}r^2_{\alpha,\beta,\gamma}(x,y)\right]\right) \\
& - \frac{1}{\sqrt{m}}\cdot\frac{1}{\sqrt{m}}\sum_{j=1}^{m}\left(\alpha r_{\alpha,\beta,\gamma}(x'_j,y'_j) + \frac{\gamma}{2}r^2_{\alpha,\beta,\gamma}(x'_j,y'_j) - \mathbb{E}_{P_X P_Y}\left[\alpha r_{\alpha,\beta,\gamma}(x,y) + \frac{\gamma}{2}r^2_{\alpha,\beta,\gamma}(x,y)\right]\right) \\
& + o_p(\frac{1}{\sqrt{n'}}).
\end{aligned}
$$

Therefore, the asymptotic Variance of $\hat{J}^{m,n}_{\mathrm{RPC}}$ is

$$
\mathrm{Var}[\hat{J}^{m,n}_{\mathrm{RPC}}] = \frac{1}{n}\mathrm{Var}_{P_{XY}}[r_{\alpha,\beta,\gamma}(x,y) - \frac{\beta}{2}r^2_{\alpha,\beta,\gamma}(x,y)] + \frac{1}{m}\mathrm{Var}_{P_X P_Y}[\alpha r_{\alpha,\beta,\gamma}(x,y) + \frac{\gamma}{2}r^2_{\alpha,\beta,\gamma}(x,y)] + o(\frac{1}{n'}).
$$

First, we look at $\mathrm{Var}_{P_{XY}}[r_{\alpha,\beta,\gamma}(x,y) - \frac{\beta}{2}r^2_{\alpha,\beta,\gamma}(x,y)]$. Since $\beta > 0$ and $-\frac{\alpha}{\gamma} \le r_{\alpha,\beta,\gamma} \le \frac{1}{\beta}$, simple calculation gives us $-\frac{2\alpha\gamma + \beta\alpha^2}{2\gamma^2} \le r_{\alpha,\beta,\gamma}(x,y) - \frac{\beta}{2}r^2_{\alpha,\beta,\gamma}(x,y) \le \frac{1}{2\beta}$. Hence,

$$
\mathrm{Var}_{P_{XY}}[r_{\alpha,\beta,\gamma}(x,y) - \frac{\beta}{2}r^2_{\alpha,\beta,\gamma}(x,y)] \le \max\left\{\left(\frac{2\alpha\gamma + \beta\alpha^2}{2\gamma^2}\right)^2, \left(\frac{1}{2\beta}\right)^2\right\}.
$$

Next, we look at $\mathrm{Var}_{P_X P_Y}[\alpha r_{\alpha,\beta,\gamma}(x,y) + \frac{\gamma}{2}r^2_{\alpha,\beta,\gamma}(x,y)]$. Since $\alpha \ge 0, \gamma > 0$ and $-\frac{\alpha}{\gamma} \le r_{\alpha,\beta,\gamma} \le \frac{1}{\beta}$, simple calculation gives us $-\frac{\alpha^2}{2\gamma} \le \alpha r_{\alpha,\beta,\gamma}(x,y) + \frac{\gamma}{2}r^2_{\alpha,\beta,\gamma}(x,y) \le \frac{2\alpha\beta + \gamma}{2\beta^2}$. Hence,

$$
\mathrm{Var}_{P_X P_Y}[\alpha r_{\alpha,\beta,\gamma}(x,y) + \frac{\gamma}{2}r^2_{\alpha,\beta,\gamma}(x,y)] \le \max\left\{\left(\frac{\alpha^2}{2\gamma}\right)^2, \left(\frac{2\alpha\beta + \gamma}{2\beta^2}\right)^2\right\}.
$$

Combining everything together, we restate the Proposition 2 in the main text:

**Proposition 4 (Asymptotic Variance of $\hat{J}^{m,n}_{\mathrm{RPC}}$)**

$$
\begin{aligned}
\mathrm{Var}[\hat{J}^{m,n}_{\mathrm{RPC}}] =& \frac{1}{n}\mathrm{Var}_{P_{XY}}[r_{\alpha,\beta,\gamma}(x,y) - \frac{\beta}{2}r^2_{\alpha,\beta,\gamma}(x,y)] + \frac{1}{m}\mathrm{Var}_{P_X P_Y}[\alpha r_{\alpha,\beta,\gamma}(x,y) + \frac{\gamma}{2}r^2_{\alpha,\beta,\gamma}(x,y)] + o(\frac{1}{n'}) \\
\le& \frac{1}{n}\max\left\{\left(\frac{2\alpha\gamma + \beta\alpha^2}{2\gamma^2}\right)^2, \left(\frac{1}{2\beta}\right)^2\right\} + \frac{1}{m}\max\left\{\left(\frac{\alpha^2}{2\gamma}\right)^2, \left(\frac{2\alpha\beta + \gamma}{2\beta^2}\right)^2\right\} + o(\frac{1}{n'})
\end{aligned}
$$

## A.5 PROOF OF PROPOSITION 2 IN THE MAIN TEXT - FROM BOUNDNESS OF $f_\theta$

As discussed in Assumption 1, for the estimation $\hat{J}_{\text{RPC}}^{m,n}$, we can bound the function $f_\theta$ in $\mathcal{F}_\Theta$ within $[-\frac{\alpha}{\gamma}, \frac{1}{\beta}]$ without losing precision. Then, re-arranging $\hat{J}_{\text{RPC}}^{m,n}$:

$$\sup_{f_\theta \in \mathcal{F}_\Theta} \frac{1}{n} \sum_{i=1}^{n} f_\theta(x_i, y_i) - \frac{1}{m} \sum_{j=1}^{m} \alpha f_\theta(x_j', y_j') - \frac{1}{n} \sum_{i=1}^{n} \frac{\beta}{2} f_\theta^2(x_i, y_i) - \frac{1}{m} \sum_{j=1}^{m} \frac{\gamma}{2} f_\theta^2(x_j', y_j')$$

$$\sup_{f_\theta \in \mathcal{F}_\Theta} \frac{1}{n} \sum_{i=1}^{n} \left( f_\theta(x_i, y_i) - \frac{\beta}{2} f_\theta^2(x_i, y_i) \right) + \frac{1}{m} \sum_{j=m}^{n} \left( \alpha f_\theta(x_j', y_j') + \frac{\gamma}{2} f_\theta^2(x_j', y_j') \right)$$

Then, since $-\frac{\alpha}{\gamma} \le f_\theta(\cdot, \cdot) \le \frac{1}{\beta}$, basic calculations give us

$$-\frac{2\alpha\gamma + \beta\alpha^2}{2\gamma^2} \le f_\theta(x_i, y_i) - \frac{\beta}{2} f_\theta^2(x_i, y_i) \le \frac{1}{2\beta} \text{ and } -\frac{\alpha^2}{2\gamma} \le \alpha f_\theta(x_j', y_j') + \frac{\gamma}{2} f_\theta^2(x_j', y_j') \le \frac{2\alpha\beta + \gamma}{2\beta^2}.$$

The resulting variances have

$$\text{Var}[f_\theta(x_i, y_i) - \frac{\beta}{2} f_\theta^2(x_i, y_i)] \le \max\left\{ \left(\frac{2\alpha\gamma + \beta\alpha^2}{2\gamma^2}\right)^2, \left(\frac{1}{2\beta}\right)^2 \right\}$$

and

$$\text{Var}[\alpha f_\theta(x_j', y_j') + \frac{\gamma}{2} f_\theta^2(x_j', y_j')] \le \max\left\{ \left(\frac{\alpha^2}{2\gamma}\right)^2, \left(\frac{2\alpha\beta + \gamma}{2\beta^2}\right)^2 \right\}.$$

Taking the mean of $m, n$ independent random variables gives the result:

**Proposition 5 (Variance of $\hat{J}_{\text{RPC}}^{m,n}$)**

$$\text{Var}[\hat{J}_{\text{RPC}}^{m,n}] \le \frac{1}{n} \max\left\{ \left(\frac{2\alpha\gamma + \beta\alpha^2}{2\gamma^2}\right)^2, \left(\frac{1}{2\beta}\right)^2 \right\} + \frac{1}{m} \max\left\{ \left(\frac{\alpha^2}{2\gamma}\right)^2, \left(\frac{2\alpha\beta + \gamma}{2\beta^2}\right)^2 \right\}.$$

## A.6 IMPLEMENTATION OF EXPERIMENTS

For visual representation learning, we follow the implementation in https://github.com/google-research/simclr. For speech representation learning, we follow the implementation in https://github.com/facebookresearch/CPC_audio. For MI estimation, we follow the implementation in https://github.com/yaohungt/Pointwise_Dependency_Neural_Estimation/tree/master/MI_Est_and_CrossModal..

## A.7 RELATIVE PREDICTIVE CODING ON VISION

The whole pipeline of pretraining contains the following steps: First, a stochastic data augmentation will transform one image sample $x_k$ to two different but correlated augmented views, $x_{2k-1}'$ and $x_{2k}'$. Then a base encoder $f(\cdot)$ implemented using ResNet (He et al., 2016) will extract representations from augmented views, creating representations $h_{2k-1}$ and $h_{2k}$. Later a small neural network $g(\cdot)$ called projection head will map $h_{2k-1}$ and $h_{2k}$ to $z_{2k-1}$ and $z_{2k}$ in a different latent space. For each minibatch of $N$ samples, there will be $2N$ views generated. For each image $x_k$ there will be one positive pair $x_{2k-1}'$ and $x_{2k}'$ and $2(N-1)$ negative samples. The RPC loss between a pair of positive views, $x_i'$ and $x_j'$ (augmented from the same image) , can be calculated by the substitution $f_\theta(x_i', x_j') = (z_i \cdot z_j)/\tau = s_{i,j}$ ($\tau$ is a hyperparameter) to the definition of RPC:

$$\ell_{i,j}^{\text{RPC}} = -(s_{i,j} - \frac{\alpha}{2(N-1)} \sum_{k=1}^{2N} \mathbf{1}_{[k \ne i]} s_{i,k} - \frac{\beta}{2} s_{i,j}^2 - \frac{\gamma}{2 \cdot 2(N-1)} \sum_{k=1}^{2N} \mathbf{1}_{[k \ne i]} s_{i,k}^2) \quad (7)$$

For losses other than RPC, a hidden normalization of $s_{i,j}$ is often required by replacing $z_i \cdot z_j$ with $(z_i \cdot z_j)/|z_i||z_j|$. CPC and WPC adopt this, while other objectives needs it to help stabilize training variance. RPC does not need this normalization.

| Confidence Interval of $J_{\mathrm{RPC}}$ and $J_{\mathrm{CPC}}$ | | | |
|---|---|---|---|
| Objective | CIFAR 10 | CIFAR 100 | ImageNet |
| $J_{\mathrm{CPC}}$ | $(91.09\%, 91.13\%)$ | $(77.11\%, 77.36\%)$ | $(73.39\%, 73.48\%)$ |
| $J_{\mathrm{RPC}}$ | $(91.16\%, 91.47\%)$ | $(77.41\%, 77.98\%)$ | $(73.92\%, 74.43\%)$ |

Table 4: Confidence Intervals of performances of $J_{\mathrm{RPC}}$ and $J_{\mathrm{CPC}}$ on CIFAR-10/-100 and ImageNet.

## A.8 CIFAR-10/-100 AND IMAGENET EXPERIMENTS DETAILS

**ImageNet**  Following the settings in (Chen et al., 2020b;c), we train the model on Cloud TPU with 128 cores, with a batch size of $4,096$ and global batch normalization [3] (Ioffe & Szegedy, 2015). Here we refer to the term batch size as the number of images (or utterances in the speech experiments) we use per GPU, while the term minibatch size refers to the number of negative samples used to calculate the objective, such as CPC or our proposed RPC. The largest model we train is a 152-layer ResNet with selective kernels (SK) (Li et al., 2019) and $2\times$ wider channels. We use the LARS optimizer (You et al., 2017) with momentum 0.9. The learning rate linearly increases for the first 20 epochs, reaching a maximum of $6.4$, then decayed with cosine decay schedule. The weight decay is $10^{-4}$. A MLP projection head $g(\cdot)$ with three layers is used on top of the ResNet encoder. Unlike Chen et al. (2020c), we do not use a memory buffer, and train the model for only 100 epochs rather than 800 epochs due to computational constraints. These two options slightly reduce CPC's performance benchmark for about $2\%$ with the exact same setting. The unsupervised pre-training is followed by a supervised fine-tuning. Following SimCLRv2 (Chen et al., 2020b;c), we fine-tune the 3-layer $g(\cdot)$ for the downstream tasks. We use learning rates 0.16 and 0.064 for standard 50-layer ResNet and larger 152-layer ResNet respectively, and weight decay and learning rate warmup are removed. Different from Chen et al. (2020c), we use a batch size of $4,096$, and we do not use global batch normalization for fine-tuning. For $J_{\mathrm{RPC}}$ we disable hidden normalization and use a temperature $\tau = 32$. For all other objectives, we use hidden normalization and $\tau = 0.1$ following previous work (Chen et al., 2020c). For relative parameters, we use $\alpha = 0.3, \beta = 0.001, \gamma = 0.1$ and $\alpha = 0.3, \beta = 0.001, \gamma = 0.005$ for ResNet-50 and ResNet-152 respectively.

**CIFAR-10/-100**  Following the settings in (Chen et al., 2020b), we train the model on a single GPU, with a batch size of $512$ and global batch normalization (Ioffe & Szegedy, 2015). We use ResNet (He et al., 2016) of depth 18 and depth 50, and does not use Selective Kernel (Li et al., 2019) or a multiplied width size. We use the LARS optimizer (You et al., 2017) with momentum 0.9. The learning rate linearly increases for the first 20 epochs, reaching a maximum of $6.4$, then decayed with cosine decay schedule. The weight decay is $10^{-4}$. A MLP projection head $g(\cdot)$ with three layers is used on top of the ResNet encoder. Unlike Chen et al. (2020c), we do not use a memory buffer. We train the model for 1000 epochs. The unsupervised pre-training is followed by a supervised fine-tuning. Following SimCLRv2 (Chen et al., 2020b;c), we fine-tune the 3-layer $g(\cdot)$ for the downstream tasks. We use learning rates 0.16 for standard 50-layer ResNet , and weight decay and learning rate warmup are removed. For $J_{\mathrm{RPC}}$ we disable hidden normalization and use a temperature $\tau = 128$. For all other objectives, we use hidden normalization and $\tau = 0.5$ following previous work (Chen et al., 2020c). For relative parameters, we use $\alpha = 1.0, \beta = 0.005,$ and $\gamma = 1.0$.

**STL-10**  We also perform the pre-training and fine-tuning on STL-10 (Coates et al., 2011) using the model proposed in Chuang et al. (2020). Chuang et al. (2020) proposed to indirectly approximate the distribution of negative samples so that the objective is *debiased*. However, their implementation of contrastive learning is consistent with Chen et al. (2020b). We use a ResNet with depth 50 as an encoder for pre-training, with Adam optimizer, learning rate 0.001 and weight decay $10^{-6}$. The temperature $\tau$ is set to 0.5 for all objectives other than $J_{\mathrm{RPC}}$, which disables hidden normalization and use $\tau = 128$. The downstream task performance increases from $83.4\%$ of $J_{\mathrm{CPC}}$ to $84.1\%$ of $J_{\mathrm{RPC}}$.

**Confidence Interval**  We also provide the confidence interval of $J_{\mathrm{RPC}}$ and $J_{\mathrm{CPC}}$ on CIFAR-10, CIFAR-100 and ImageNet, using ResNet-18, ResNet-18 and ResNet-50 respectively (95% confi-

---

[3]For WPC (Ozair et al., 2019), the global batch normalization during pretraining is disabled since we enforce 1-Lipschitz by gradient penalty (Gulrajani et al., 2017).

dence level is chosen) in Table 4. Both CPC and RPC use the same experimental settings throughout this paper. Here we use the relative parameters ($\alpha = 1.0, \beta = 0.005, \gamma = 1.0$) in $J_{\text{RPC}}$ which gives the best performance on CIFAR-10. The confidence intervals of CPC do not overlap with the confidence intervals of RPC, which means the difference of the downstream task performance between RPC and CPC is statistically significant.

## A.9 RELATIVE PREDICTIVE CODING ON SPEECH

For speech representation learning, we adopt the general architecture from Oord et al. (2018). Given an input signal $\boldsymbol{x}_{1:T}$ with $T$ time steps, we first pass it through an encoder $\phi_\theta$ parametrized by $\theta$ to produce a sequence of hidden representations $\{\boldsymbol{h}_{1:T}\}$ where $\boldsymbol{h}_t = \phi_\theta(\boldsymbol{x}_t)$. After that, we obtain the contextual representation $\boldsymbol{c}_t$ at time step $t$ with a sequential model $\psi_\rho$ parametrized by $\rho$: $\boldsymbol{c}_t = \psi_\rho(\boldsymbol{h}_1, \ldots, \boldsymbol{h}_t)$, where $\boldsymbol{c}_t$ contains context information before time step $t$. For unsupervised pre-training, we use a multi-layer convolutional network as the encoder $\phi_\theta$, and an LSTM with hidden dimension 256 as the sequential model $\psi_\rho$. Here, the contrastiveness is between the positive pair $(\boldsymbol{h}_{t+k}, \boldsymbol{c}_t)$ where $k$ is the number of time steps ahead, and the negative pairs $(\boldsymbol{h}_i, \boldsymbol{c}_t)$, where $\boldsymbol{h}_i$ is randomly sampled from $\mathcal{N}$, a batch of hidden representation of signals assumed to be unrelated to $\boldsymbol{c}_t$. The scoring function $f$ based on Equation 2 at step $t$ and look-ahead $k$ will be $f_k = f_k(\boldsymbol{h}, \boldsymbol{c}_t) = \exp((\boldsymbol{h})^\top \boldsymbol{W}_k \boldsymbol{c}_t)$, where $\boldsymbol{W}_k$ is a learnable linear transformation defined separately for each $k \in \{1, ..., K\}$ and $K$ is predetermined as 12 time steps. The loss in Equation 2 will then be formulated as:

$$\ell_{t,k}^{\text{RPC}} = -(f_k(\boldsymbol{h}_{t+k}, \boldsymbol{c}_t) - \frac{\alpha}{|\mathcal{N}|} \sum_{\boldsymbol{h}_i \in \mathcal{N}} f_k(\boldsymbol{h}_i, \boldsymbol{c}_t) - \frac{\beta}{2} f_k^2(\boldsymbol{h}_{t+k}, \boldsymbol{c}_t) - \frac{\gamma}{2|\mathcal{N}|} \sum_{\boldsymbol{h}_i \in \mathcal{N}} f_k^2(\boldsymbol{h}_i, \boldsymbol{c}_t)) \quad (8)$$

We use the following relative parameters: $\alpha = 1, \beta = 0.25$, and $\gamma = 1$, and we use the temperature $\tau = 16$ for $J_{\text{RPC}}$. For $J_{\text{CPC}}$ we follow the original implementation which sets $\tau = 1$. We fix all other experimental setups, including architecture, learning rate, and optimizer. As shown in Table 3, $J_{\text{RPC}}$ has better downstream task performance, and is closer to the performance from a fully supervised model.

## A.10 EMPIRICAL OBSERVATIONS ON VARIANCE AND MINIBATCH SIZE

**Variance Experiment Setup** We perform the variance comparison of $J_{\text{DV}}$, $J_{\text{NWJ}}$ and the proposed $J_{\text{RPC}}$. The empirical experiments are performed using SimCLRv2 (Chen et al., 2020c) on CIFAR-10 dataset. We use a ResNet of depth 18, with batch size of 512. We train each objective with 30K training steps and record their value. In Figure 1, we use a temperature $\tau = 128$ for all objectives. Unlike other experiments, where hidden normalization is applied to other objectives, we remove hidden normarlization for all objectives due to the reality that objectives after normalization does not reflect their original values. From Figure 1, $J_{\text{RPC}}$ enjoys lower variance and more stable training compared to $J_{\text{DV}}$ and $J_{\text{NWJ}}$.

**Minibatch Size Experimental Setup** We perform experiments on the effect of batch size on downstream performances for different objective. The experiments are performed using SimCLRv2 (Chen et al., 2020c) on CIFAR-10 dataset, as well as the model from Rivière et al. (2020) on LibriSpeech-100h dataset (Panayotov et al., 2015). For vision task, we use the default temperature $\tau = 0.5$ from Chen et al. (2020c) and hidden normalization mentioned in Section 3 for $J_{\text{CPC}}$. For $J_{\text{RPC}}$ in vision and speech tasks we use a temperature of $\tau = 128$ and $\tau = 16$ respectively, both without hidden normalization.

## A.11 MUTUAL INFORMATION ESTIMATION

Our method is compared with baseline methods CPC (Oord et al., 2018), NWJ (Nguyen et al., 2010), JSD (Nowozin et al., 2016), and SMILE (Song & Ermon, 2019). All the approaches consider the same design of $f(x, y)$, which is a 3-layer neural network taking concatenated $(x, y)$ as the input. We also fix the learning rate, the optimizer, and the minibatch size across all the estimators for a fair comparison.

We present results of mutual information by Relative Predictive Coding using different sets of relative parameters in Figure 4. In the first row, we set $\beta = 10^{-3}, \gamma = 1$, and experiment with different

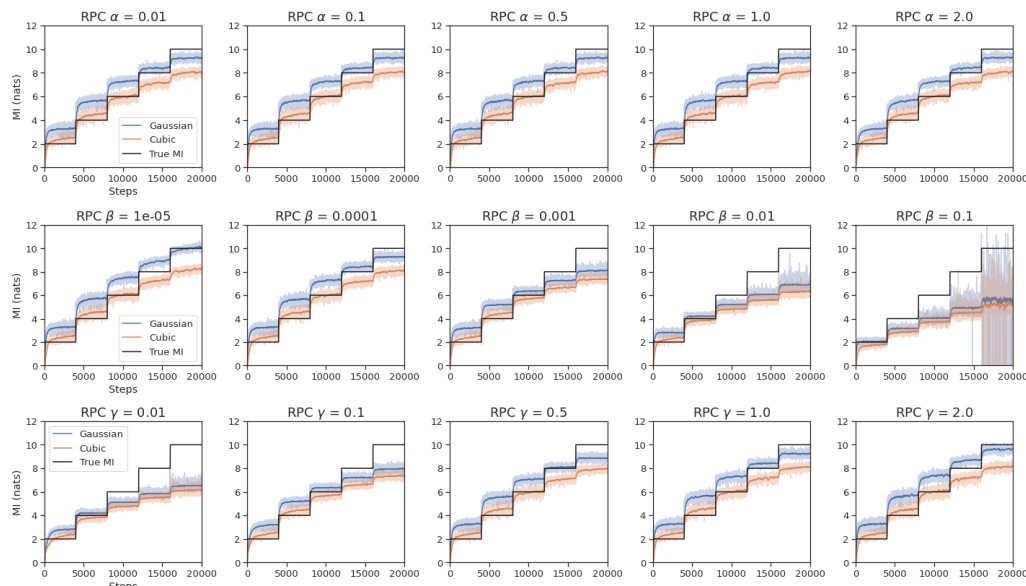

Figure 4: Mutual information estimation by RPC performed on 20-d correlated Gaussian distribution, with different sets of relative parameters.

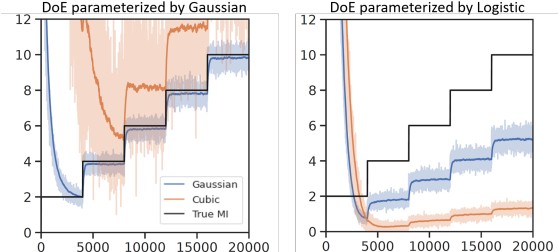

Figure 5: Mutual information estimation by DoE performed on 20-d correlated Gaussian distribution. The figure on the left shows parametrization under Gaussian (correctly specified), and the figure on the right shows parametrization under Logistic (mis-specified).

$\alpha$ values. In the second row, we set $\alpha = 1$, $\gamma = 1$ and in the last row we set $\alpha = 1$, $\beta = 10^{-3}$. From the figure, a small $\beta$ around $10^{-3}$ and a large $\gamma$ around 1.0 is crucial for an estimation that is relatively low bias and low variance. This conclusion is consistent with Section 3 in the main text.

We also performed comparison between $J_{\text{RPC}}$ and Difference of Entropies (DoE) (McAllester & Stratos, 2020). We performed two sets of experiments: in the first set of experiments we compare $J_{\text{RPC}}$ and DoE when MI is large ($> 100$ nats), while in the second set of experiments we compare $J_{\text{RPC}}$ and DoE using the setup in this section (MI $< 12$ nats and MI increases by 2 per 4k training steps). On the one hand, when MI is large ($> 100$ nats), we acknowledge that DoE is performing well on MI estimation, compared to $J_{\text{RPC}}$ which only estimates the MI around 20. This analysis is based on the code from https://github.com/karlstratos/doe. On the other hand, when the true MI is small, the DoE method is more unstable than $J_{\text{RPC}}$, as shown in Figure 5. Figure 5 illustrates the results of the DoE method when the distribution is isotropic Gaussian (correctly specified) or Logistic (mis-specified). Figure 3 only shows the results using Gaussian.

