# OpenReview forum: "Self-supervised Representation Learning with Relative Predictive Coding"
_ICLR.cc/2021/Conference — ICLR 2021 Poster_

### Official Review · AnonReviewer3 · 2020-10-27
**Interesting direciton towards stable self-supervised training objectives**

**Rating:** 7
**Confidence:** 3

**Review:**

Authors propose RPC (Relative Predictive Coding), which is supposed to improve training stability (with chi-squared distance based regularization), minibatch size sensitivity (avoid sampling large batches), and downstream task performance (show generalization).
Authors discuss estimation of MI and/or probability ratio (of related divided by unrelated egs). Proposed solution stably estimates this. Experiments are convincing. It is a good direction in self-supervised training with convenient training schemes.
Some weaknesses:
1. Fix alpha=0 and find ratio or values of beta and gamma which gives maximum performance. It would be interesting since low value of alpha is also giving good performance.
2. While discussing sensitivity to batch size, larger batch sizes should be tried since it is discussed in the initial part of paper that SimCLRv2 requires huge batch size.
3. Since the proposal is generic, can authors give a word on using this on something other than SimCLRv2?
4. Rather than reporting specific values of alpha, beta, gamma (the proposed "relative parameters"), if results can be reported in graph format, it would be vastly more helpful. For e.g. fix alpha=0.001 and x and y axis of plot could be other two relative parameters. (this is related to my point 1)

---

> ### Author Response · Authors · 2020-11-17
> **Response**
>
> [Additional experiments on more sets of relative parameters]
>
> We provide the experimental results in the revised manuscript and highlight them in red in Section 3.4. The experiments are focusing on 1) providing the results when $\alpha=0$ and 2) changing the table to the graph format of the results by fixing $\alpha$ and changing $\beta$ and $\gamma$. Below are the discussions.
>
> We study the effect of different combinations of relative parameters in $J_{\rm RPC}$ by comparing downstream performances on visual object recognition. We train SimCLRv2 on CIFAR-10 with different combinations of $\alpha, \beta$ and $\gamma$ in $J_{\rm RPC}$ and fix all other experimental settings. We choose $\alpha = \{ 0, 0.001, \text{ and }1.0 \}; \beta = \{ 0, 0.001, \text{ and }1.0 \}; \gamma = \{ 0, 0.001, \text{ and } 1.0 \ }$, then we report the best performances under each combination of $\alpha, \beta$, and $\gamma$. From Figure 2 in the paper, we first observe that $\alpha > 0$ has better downstream performance than $\alpha=0$ when $\beta$ and $\gamma$ are fixed. This observation is as expected, since $\alpha >0$ encourages representations of related and unrelated samples to be pushed away. Then, we find that a small but nonzero $\beta$ ($\beta = 0.001$) and a large $\gamma$ ($\gamma = 1.0$) give the best performance compared to other combinations. Since $\beta$ and $\gamma$ serve as the coefficients of $\ell_2$ regularization, the results imply that the regularization is a strong and sensitive factor that will influence the performance. In conclusion, we find empirically that a non-zero $\alpha$, a small $\beta$ and a large $\gamma$ will lead to the optimal representation for the downstream task.
>
> [Remarks on the larger minibatch size]
>
> During the rebuttal period, we no longer have access to Google TPU cloud resources on which we perform all our large minibatch size experiments. For this paper, large minibatch size experiments can only be performed on TPUs, since consumer-level GPUs do not have enough memory to hold a large minibatch and the model at the same time. To be particular, we have no computational resources to perform a comparison between RPC and CPC under a range of large minibatch sizes (for example, when minibatch sizes are 2,048, 4,096, 8,192, and 1,6384, which is the minibatch sizes used in SimCLRv2 [2]) on ImageNet. As a compromise, in the paper, we provided minibatch sensitivity experiments on the CIFAR-10 dataset. The largest minibatch size we considered is 512, which is the minibatch size chosen in SimCLR [1] and SimCLRv2 [2] for CIFAR-10.
>
> [Remarks on using other architectures]
>
> We have also performed experiments using the architecture in AMDIM [3], and the results and conclusions are the same (i.e., RPC has better downstream performance, less minibatch size sensitivity and better training stability) compared to using the architectures in SimCLR [1] and SimCLRv2 [2]. We decided to report the results using the architecture in SimCLRv2 due to its better performance.
>
>
> [1] Chen et al., "A simple framework for contrastive learning of visual representations", ICML 2020.
>
> [2] Chen et al., “Big Self-Supervised Models are Strong Semi-Supervised Learners”, NeurIPS 2020.
>
> [3] Bachman et al., “Learning representations by maximizing mutual information across views”, NeurIPS 2019.

---

### Official Review · AnonReviewer2 · 2020-10-28
**Solid contribution to contrastive representation learning**

**Rating:** 8
**Confidence:** 4

**Review:**

The authors provide a clear review of different divergences used in contrastive learning and their relative strengths and weaknesses in terms of training stability, minibatch size dependence, and usefulness on downstream tasks. This motivates the need for a new divergence which they introduce based upon chi-squared divergence.

They provide strong empirical and theoretical support for the new divergence, with extensive experiments on large-scale image and speech classification tasks. They also perform comparison studies across batch size and training stability that support their earlier arguments, and a hyperparameter sweep across term weights to make it clearer how to tune them in later work. Further, they demonstrate the decreased bias and variance in MI estimation experiments.

The paper is well-written, and provides helpful context to not just motivate the value of the new technique, but quantitatively and qualitatively contrast with existing techniques that helps inform the reader about the broader field.

---

> ### Author Response · Authors · 2020-11-17
> **Response**
>
> Thank you for the valuable comments. We appreciate the acknowledgment of our motivation, the theoretical support of the new divergence, as well as the quantitative and qualitative comparisons with existing techniques.

---

### Official Review · AnonReviewer1 · 2020-10-29
**This paper presents a new contrastive representation objective that has good training stability, minibatch size sensitivity, and downstream task performance.**

**Rating:** 6
**Confidence:** 4

**Review:**

This paper presents a new contrastive representation objective that has good training stability, minibatch size sensitivity, and downstream task performance. This objective is a generalization of Chi-square divergence, the optimal solution is the density ratio of joint distribution and product of marginal distributions, the estimation is consistent and its variance goes to 0 as sample size goes to infinite, so the paper is theoretical sound. The authors conduct comprehensive experiments to show that the training based on this objective is stable, not sensitive to batch size and leads to good downstream task performance  in vision and phoneme and speaker classification.

However the theoretical results don't provide any clue to explain why this estimation leads to better downstream task performance, where in mutual information (MI) estimator it is easy to understand since MI is a good measure of dependency of two random variables. Section 3.5 states the relation to MI estimation, but it is just a plug-in method that won't be able to say anything on the goodness.

Furthermore the paper misses an important reference, David McAllester and Karl Stratos, Formal Limitations on the Measurement of Mutual Information, AISTATS 2020, where David and Karl propose a method called DoE that has a good estimation of mutual information even when the mutual information is large. So I think the authors should compare RPC with DoE in the synthetic data experiment in Section 3.5 when the mutual information is large, say 100, instead of 10. In the case of large MI, all other methods fail to provide a good estimation.

From my experience, when CPC is applied to ASR, the batch size is not a sensitive factor to WER results.

In the notation and In section 2.1, second line from bottom, I don't think it is appropriate to use joint distribution for a related or positive pair, product of marginal distributions for an un-related or negative pair.  It is the density ratio matters. For positive pair (X,Y), its MI is large, and for negative pair (X,Y), its MI close to 0.

In the proof of Lemma 5, there is a typo in the second line from bottom, the second term should be expection of P(x)P(Y), not P(X,Y).

---

> ### Author Response · Authors · 2020-11-17
> **Response**
>
> [Remarks on the downstream task performance]
>
> Previous works inspired by mutual information estimators on contrastive self-supervised representation learning maximize the dependency (via KL divergence) of representations learned from related data. The intuition follows by making the model have a discriminative power to “pull over” the related features and “push away” the unrelated features. Our proposed RPC has the same goal as this line of works, by changing the divergence in the dependency measurement from the KL to the Chi-squared divergence.
>
> We would like to emphasize that the goal of Section 3.5 is NOT to back up the claim on a good downstream performance using the proposed RPC method. Instead, we relate our method to mutual information estimation in Section 3.5 because the density ratio (i.e., p(x,y)/p(x)p(y)) is an intermediate product in our proposed RPC method. As pointed out in prior work [1]: theoretically and empirically, if the density ratio can be properly estimated, then plugging-in the estimated density ratio to estimate the mutual information leads to low bias and variance. We hence disagree with the claim that “just a plug-in method that won't be able to say anything on the goodness” on the mutual information estimation.
>
> [Discussion with related work DoE method [2]]
>
> We thank the reviewer for suggesting a comparison and discussion with the relevant work [2]. We would like to emphasize that the bulk of our efforts and our primary contribution is to present RPC as a new self-supervised representation learning approach, instead of constructing a dedicated MI estimator. The mutual information estimation is a byproduct of our work. Nonetheless, we are happy to provide an experimental comparison with the DoE method [2]. The experimental results are highlighted in red in the revised manuscript in Section 3.5 and Appendix A.11. In the following, we provide a discussion on the new experimental results.
>
> We perform two sets of experiments to compare RPC v.s. DoE method: the first set considers the case when MI is large (>100 nats) and the second set considers our experimental setup in Section 3.5 (MI < 12 nats and MI increases by 2 per 4k training steps). On the one hand, we acknowledge that the DoE method is performing well on MI estimation when MI is very large (> 100 nats), compared to RPC which only estimates the MI around 20. On the other hand, when the true MI is small, RPC achieves a better estimation than the DoE method and the trend is more obvious in the cubic task.
>
> [Batch size in ASR results using CPC in terms of WER]
>
> We have addressed the concern in the revised manuscript and highlighted it in red in Appendix A.8.
>
> In speech experiments, to keep it consistent throughout the whole paper, we refer to the term batch size as the number of utterances we use per GPU, while the term minibatch size refers to the number of negative samples used to calculate the loss objective, such as CPC or our proposed RPC. The minibatch size (number of negative samples) is of the paper’s interest because CPC uses negative samples and previous work [3] shows that the performance of CPC is sensitive to the number of negative samples, i.e. minibatch size. The experimental results in Figure 1 (c) shows performances under different numbers of negative samples (i.e., the minibatch size), but not the number of utterances (i.e., the batch size).
>
> [Remarks on the notion of the positive and the negative pair]
>
> We appreciate the suggestion from the reviewer and have made the following modifications. We include the change in the revised manuscript and highlight them in red in Section 2.1.
>
> Contrastive representation learning encourages the contrastiveness between the positive and negative pairs of the representations from the related data X and Y. Specifically when sampling a pair of representations (x,y) from their joint distribution ((x,y) ~ P_{XY}), this pair is defined as a positive pair; when sampling from the product of marginals ((x,y) ~ P_XP_Y), this pair is defined as a negative pair.
>
> [Typo in the proof]
>
> We have fixed the typo and highlight the change in red in Appendix A.3.1 in the revised manuscript.
>
> [1] Tsai et al., “Neural Methods for Point-wise Dependency Estimation”, NeurIPS 2020.
>
> [2] McAllester et al., “Formal limitations on the measurement of mutual information”, AISTATS 2020.
>
> [3] Ozair et al., “Wasserstein Dependency Measure for Representation Learning”, NeurIPS 2019

---

> > ### Comment · AnonReviewer1 · 2020-11-24
> > **Official Blind Review #1**
> >
> > I think the rebuttal addresses most of my concerns.
> >
> > My only concern is on Figures 3-5, these figures compares RPC with other MI related methods, however none illustrates the problems of using these methods except DoE when MI is large. I think it's better add a figure in the main body that is similar to Figure 3 but the MIs are 10, 20, ...., 100.
> >
> > From theoretical side, if Chi-squared divergence is able to estimate the density ratio properly, and plugging-in method works, then should f-divergence work in general? This is unclear.

---

> > > ### Author Response · Authors · 2020-11-24
> > > **Response to the following review**
> > >
> > > We thank you for the follow-up reviews!
> > >
> > > [More comparisons with the DoE method]
> > > Since the rebuttal period ends today, we will try our best to provide the plot with large MIs (10, 20, ..., 100) comparing our method with the DoE method. If we're not able to provide the new results today, we will add it to the revision.
> > >
> > > [Will the plugging-in density ratio methods work for f-divergence in general from a theoretical perspective?]
> > > We thank the reviewer for pointing out this concern. The rationale behinds that the plugging-in the density ratio will work for the mutual information is provided in the prior work [1], and we are happy to discuss the extension to a general f-divergence measurement.
> > >
> > > First of all, we note that we need a couple of assumptions to claim that plugging-in the density ratio will work for estimating mutual information. The assumptions are 1) the boundness of the density ratio, 2) the log-smoothness of the density ratio and 3) neural network universal approximation lemma. To extend the proof from mutual information to general f-divergence, we simply need to alter the assumption from the second assumption (the log-smoothness assumption). For example, if considering the f-divergence as the chi-square divergence, then we need the smoothness of the network parameters on the squared density-ratio function instead of the log density-ratio function.
> > >
> > > Although being happy to discuss this extension, we believe this question is out of the scope of our paper. Estimating general f-divergence properly and efficiently is itself a big challenge, while the main goal of our paper is providing a new objective for self-supervised representation learning.
> > >
> > > [1] Tsai et al., “Neural Methods for Point-wise Dependency Estimation”, NeurIPS 2020.

---

### Official Review · AnonReviewer4 · 2020-10-29
**Incremental contribution to contrastive learning objective**

**Rating:** 6
**Confidence:** 3

**Review:**

This paper proposes a new objective for self-supervised contrastive learning. In the general framework proposed by Tsai et al. (2020b), the proposed method boils down to using a divergence related to $\chi^2$-divergence. Compared to other objectives for contrastive learning, the authors illustrate the advantages of the proposed one in training stability (or easiness to train), sensitivity to batch size, and downstream task performance. However, introducing three new hyperparameters is a cause of concern since they make it more difficult to select optimal hyperparameters. Also, some important details of the experiments are missing. For example, how many runs to obtain the results shown in Tables 2 & 3? What's the confidence interval on the results? Any test to establish the statistical significance? What are the settings for supervised training? When the authors compare the results among different methods, did they select the optimal hyperparameters (e.g., learning rate) separately for each method?

---

> ### Author Response · Authors · 2020-11-17
> **Response**
>
> [More statistics on the experimental results]
>
> In Appendix A.8 we mention that we run the ImageNet experiments for 100 epochs and CIFAR-10/100 for 1000 epochs. We report average performance over 3 runs. We report the confidence interval and highlight them in red in Appendix A.8.
>
> We provide the confidence interval of RPC and CPC on CIFAR-10, CIFAR-100 and ImageNet, using ResNet-18, ResNet-18 and ResNet-50 respectively (95% confidence level is chosen):
>
>
> ║ Objective ║    CIFAR-10    ║    CIFAR-100   ║    ImageNet    ║
>
>
> ║    CPC    ║ (91.09, 91.13) ║ (77.11, 77.36) ║ (73.39, 73.48) ║
>
>
> ║    RPC    ║ (91.16, 91.47) ║ (77.41, 77.98) ║ (73.92, 74.43) ║
>
>
> Both CPC and RPC use the default parameters provided in our paper. The confidence intervals of CPC do not overlap with the confidence intervals of RPC, which means the difference of the downstream task performance between RPC and CPC is statistically significant.
>
> [Supervised Setting]
>
> For the vision experiments, we directly report the performance of supervised learning from SimCLRv2 paper [1]. For each row in Table 2 in our manuscript, the supervised model uses the same network structure as the corresponding self-supervised model (for example, ResNet-152 is used for the supervised model as well as the self-supervised model on ImageNet). All the supervised models are trained from scratch with 90 epochs. For the speech experiments, we directly report the numbers from the prior work [2], where the supervised model and the self-supervised model also have the same network design.
>
> [Optimal hyperparameters (e.g., learning rate) selection]
>
> We use the same set of hyperparameters when comparing prior methods with the proposed RPC. The only difference among different methods is the design of the objective function. We select these hyperparameters based on the best performance on the CPC objective (not our RPC objective). In Appendix A.8, we provided the hyperparameters we used, including learning rate, weight decay, momentum, etc.
>
> [1] Chen et. al., “Big Self-Supervised Models are Strong Semi-Supervised Learners”, NeurIPS 2020.
>
> [2] Oord et. al., “Representation Learning with Contrastive Predictive Coding”, 2018.

---

### Author Response · Authors · 2020-11-17
**General Response**

We thank all reviewers for the thoughtful feedback. We thank the reviewers for acknowledging the motivation as clear (R1-R4), the contribution as solid (R2), the direction as good (R3), and the theoretical (R1, R2) and experimental (R1, R2, R3) parts as convincing and comprehensive. We have addressed the concerns from the reviewers below and provided the suggested experimental results in the revised manuscript.

---

### Decision · Program_Chairs · 2021-01-07
**Final Decision**

**Decision:**

Accept (Poster)

**Comment:**

The paper presents new contrastive based self-supervised objective based on Chi squared divergence that helps with mini batch sensitivity, training stability and improved downstream performance.
An accept.